# *FastDoc*: Domain-Specific Fast Continual Pre-training Technique using Document-Level Metadata and Taxonomy

**Abhilash Nandy**                                                     *nandyabhilash@kgpian.iitkgp.ac.in*
*Department of Computer Science*
*Indian Institute of Technology Kharagpur*

**Manav Nitin Kapadnis**                                                 *mkapadni@andrew.cmu.edu*
*School of Computer Science*
*Carnegie Mellon University*

**Sohan Patnaik**                                                      *sohanpatnaik106@iitkgp.ac.in*
*Department of Mechanical Engineering*
*Indian Institute of Technology Kharagpur*

**Yash Parag Butala**                                                        *ypb@andrew.cmu.edu*
*School of Computer Science*
*Carnegie Mellon University*

**Pawan Goyal**                                                          *pawang@cse.iitkgp.ac.in*
*Department of Computer Science*
*Indian Institute of Technology Kharagpur*

**Niloy Ganguly**                                                         *niloy@cse.iitkgp.ac.in*
*Department of Computer Science*
*Indian Institute of Technology Kharagpur*

**Reviewed on OpenReview:** `https://openreview.net/forum?id=RA4yRhjoXw`

## Abstract

In this paper, we propose ***FastDoc*** (**Fast** Continual Pre-training Technique using **Doc**ument Level Metadata and Taxonomy), a novel, compute-efficient framework that utilizes Document metadata and Domain-Specific Taxonomy as supervision signals to continually pre-train transformer encoder on a domain-specific corpus. The main innovation is that during domain-specific pretraining, an open-domain encoder is continually pre-trained using sentence-level embeddings as inputs (to accommodate long documents), however, fine-tuning is done with token-level embeddings as inputs to this encoder. We perform such domain-specific pre-training on three different domains namely customer support, scientific, and legal domains, and compare performance on 6 different downstream tasks and 9 different datasets. The novel use of document-level supervision along with sentence-level embedding input for pre-training reduces pre-training compute by around $1,000$, $4,500$, and $500$ times compared to MLM and/or NSP in Customer Support, Scientific, and Legal Domains, respectively[1]. The reduced training time does not lead to a deterioration in performance. In fact we show that ***FastDoc*** either outperforms or performs on par with several competitive transformer-based baselines in terms of character-level F1 scores and other automated metrics in the Customer Support, Scientific, and Legal Domains. Moreover, reduced training aids in mitigating the risk of catastrophic forgetting. Thus, unlike baselines, ***FastDoc*** shows a negligible drop in performance on open domain.

---

[1]Code and datasets are available at `https://github.com/manavkapadnis/FastDoc-Fast-Pre-training-Technique/`

# 1 Introduction

In present times, continual pre-training (Arumae et al., 2020; Gururangan et al., 2020) on unlabelled, domain-specific text corpora (such as PubMed articles in medical domain, research papers in Scientific Domain, E-Manuals in Customer Support Domain, etc.) has emerged as an important training strategy in NLP to enable open-domain transformer-based language models perform various downstream NLP tasks such as Question Answering (QA), Named Entity Recognition (NER), Natural Language Inference (NLI), etc. on domain-specific datasets (Hendrycks et al., 2021; Beltagy et al., 2019; Nandy et al., 2021). Most of the pre-training strategies involve variants of Masked Language Modelling (MLM) (Liu et al., 2019), Next Sentence Prediction (NSP) (Devlin et al., 2019), Sentence Order Prediction (SOP) (Lan et al., 2019), etc. that use local sentence/span-level contexts as supervision signals. However, such methods require a lot of pre-training data and compute. For instance - pre-training of $BERT_{BASE}$ architecture on a 3.17 billion word corpus was performed on 8 GPUs for around 40 days to obtain SciBERT (Beltagy et al., 2019).

MLM-style domain-specific pre-training makes an implicit assumption that the constituent documents are independent of each other, which may not be true always. Documents from a particular domain (e.g., customer support, scientific papers, legal proceedings, etc.) may be categorized into different groups by experts in that area, each group containing similar documents. This information is generally stored as either 'metadata' of the document (Borchert et al., 2020; 2022; Lipscomb, 2000), or in terms of a 'taxonomy' (Margiotta et al., 2022; Karamanolakis et al., 2020) of documents. For example, E-manuals of different versions of a cell phone series are very similar, scientific articles written on a particular topic (e.g., pre-training) follow a certain type of taxonomy, legal proceedings on related crimes are similar. While few models such as LinkBERT (Yasunaga et al., 2022), MetricBERT (Malkiel et al., 2022), etc. have used document metadata as an additional signal, no work to the best of our knowledge has *singularly* leveraged taxonomy-based information [2].

Contrarily, in this paper, we completely replace the local context-based supervision (MLM, NSP, etc.) during pre-training with (a). document similarity learning task using the available domain-specific metadata (through a triplet network), and (b). hierarchical classification task that predicts the hierarchical categories corresponding to the domain-specific taxonomy in a supervised manner.

However, to leverage document-level supervision, a robust encoding of documents is required. We use a hierarchical architecture (Zhang et al., 2019) and propose various innovations (see Figure 1) - (a). We initialize the lower-level encoder using a pre-trained sentence transformer (sBERT/sRoBERTa (Reimers & Gurevych, 2019)) and freeze its weights. We then initialize the higher-level encoder using pre-trained BERT/RoBERTa encoder, which now operates with a sentence embedding input, received via the lower-level encoder. This design choice (inspired by works that initialize a larger encoder through a smaller pre-trained encoder - e.g., Bert2BERT (Chen et al., 2022)) helps us to directly work with sentence embeddings as inputs which in turn enables much larger contexts in a single input, and decreases the required pre-training compute by a huge margin. (b). After pre-training, we use only the higher-level encoder for downstream sentence and token-level tasks. As the higher-level encoder was originally pre-trained with token embedding inputs, it can still be fine-tuned with token embedding inputs. We conduct various experiments to analyze this very interesting and surprising aspect of interoperability of token and sentence embedding inputs.

Using these ideas, we propose ***FastDoc*** pre-training framework, and apply it to varied NLP tasks across three disparate domains - **Customer Support**, **Scientific Papers**, and **Legal Domain**, to evaluate the generalizability of ***FastDoc*** across multiple domains[3]. Customer Support requires answering consumer queries related to device maintenance, troubleshooting, etc., and hence, we apply ***FastDoc*** on two **Question Answering** tasks. In the domain of scientific papers, we focus on tasks such as extracting important scientific keywords (Li et al., 2016; Kim et al., 2004; Doğan et al., 2014), extracting the type of relation between such keywords (Kringelum et al., 2016; Luan et al., 2018), as well as classifying citation intents (Cohan et al.,

---

[2] Detailed Prior Art is described ***in Section 8 of Appendix.***

[3] Continually Pre-training a single model across domains does not give good performance in all domains. That is why there are works for developing models for a particular domain, such as BioBERT Lee et al. (2020), SciBERT Beltagy et al. (2019), EManuals-BERT Nandy et al. (2021), Legal-BERT Chalkidis et al. (2020), FinBERT Huang et al. (2023)

2019). In the legal domain, we focus on the task of automating *contract review* (Hendrycks et al., 2021), which involves finding key clauses in legal contracts.

We show that **FastDoc** drastically reduces (order of 500x) pre-training compute across domains while still achieving comparable to modestly better performance in downstream tasks. We further show that the result holds even when we increase model size and consider situations where document metadata and taxonomy may not be explicitly available. We also show that the frugal pre-training helps **FastDoc** resist catastrophic forgetting so very common when transformers undergo continual in-domain pre-training (Gururangan et al., 2020; Arumae et al., 2020).

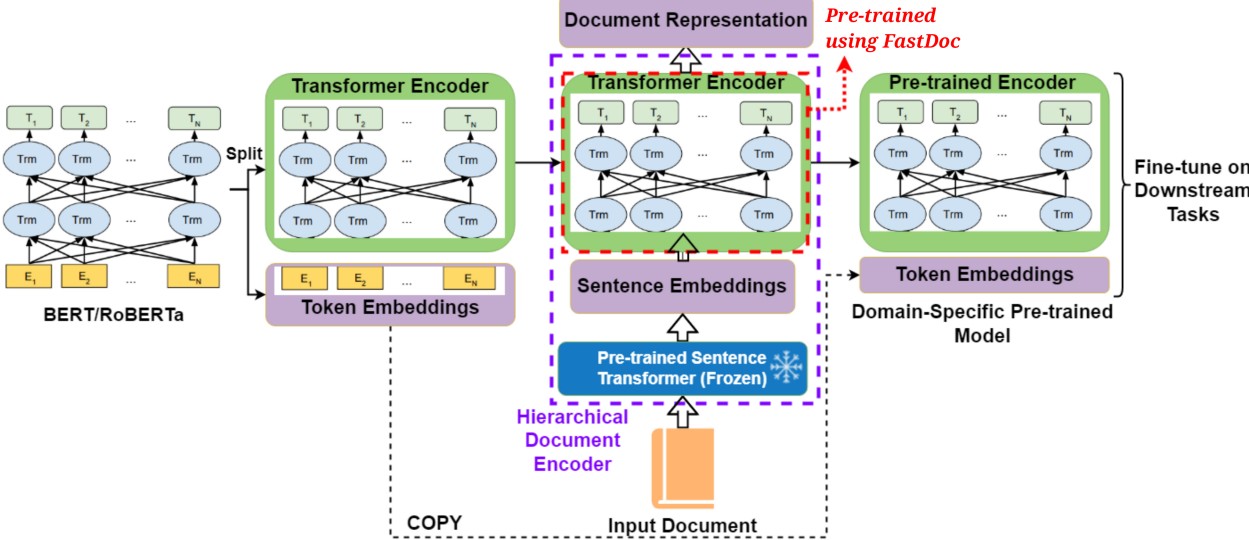

Figure 1: End-to-end training pipeline using **FastDoc**

## 2 **FastDoc** Framework

The aim of **FastDoc** is to learn robust representations for documents (in specialized domains) using potent document-level supervision signals. We treat a document as a sequence of sentences and provide pre-trained sentence embeddings as input. This, in turn, helps in accommodating documents that contain more than 512 tokens even using a standard $\text{BERT}_{\text{BASE}}$ /$\text{RoBERTa}_{\text{BASE}}$ encoder (e.g. from Figure 3 **in Section B of Appendix**, we observe that using sentences as inputs enable coverage of around 50% more documents than when tokens are inputs). We train the network with two losses. (a). *The first loss* is a contrastive or triplet loss based on the similarity or dissimilarity of a document with a pair of documents; (b). *The second loss* is a supervised loss derived while classifying a document to a domain-specific taxonomy.

Figure 1 depicts the end-to-end training pipeline using the proposed **FastDoc** architecture (detailed pre-training architecture shown in Figure 4 **in Section B of Appendix**). Typically a hierarchical document encoder like HiBERT (Zhang et al., 2019) would be a suitable model for encoding documents. It has a lower-level encoder with token inputs and a higher-level encoder with sentence-level inputs. In general, during pre-training, both these encoders need to be tuned (which is computationally expensive) and only the lower-level encoder is utilized for downstream sentence and token-level tasks such as QA, Relation Classification, NER, etc. However, we propose a different, compute-efficient method. The steps in our pipeline are - (a). The pipeline starts using an open-domain pre-trained transformer model (e.g. BERT (Devlin et al., 2019)/RoBERTa (Liu et al., 2019)) for fast convergence in domain-specific scenarios. (b). Its transformer layers excluding the input token embedding layer are used to initialize the higher-level encoder, while the lower-level encoder is a frozen sBERT/sRoBERTa. The Document representation from this document encoder is obtained by averaging the output context-aware sentence representations from the higher-level encoder. (c). The higher-level encoder is (further) pre-trained with document-level supervision

using the proposed **FastDoc** Framework on domain-specific documents. (d). Finally, only this higher-level encoder is fine-tuned on downstream tasks, with input token embeddings copied from the open-domain model.

Our specific design choices help in the following manner - (a) Freezing the sentence embeddings while training the encoder with document-level loss helps in achieving fast pre-training, (b) While a hierarchical encoder could also have used the document-level loss, the lower-level encoder using token inputs would be directly used for fine-tuning, but this encoder would learn less robust pre-training task-specific, semantic features as compared to the higher-level encoder (Tenney et al., 2019; van Aken et al., 2019). Our design trains the higher-level encoder to make the best use of pre-training loss. Next, we describe the pre-training loss functions in great detail. The inter-operability of input token and sentence embeddings is reasoned via several experiments in Section 7.4 and ***Section G.4 of Appendix.***

### Contrastive Learning using document similarity labels.

We use a Triplet Network (Cohan et al., 2020), where three documents serve as input for three document encoders, the first (anchor) and second (positive) documents being similar, and the first and third (negative) documents being dissimilar (based on metadata). The encoders have hard parameter sharing (Caruana, 1993). The three encoded representations are used to formulate a triplet margin loss function, denoted by $\mathcal{L}_t$. Mathematically,

$$\mathcal{L}_t(D_1, D_2, D_3) = max\{d(D_1, D_2) - d(D_1, D_3) + 1, 0\} \tag{1}$$

where $D_1, D_2, D_3$ refer to the document representations of documents, and $d(.,.)$ represents the L2 norm distance. We use a unit margin in accordance with prior art using the same or similar contrastive loss functions (Oh Song et al., 2016; Weinberger & Saul, 2009).

Note that we do not use NT-Xent (normalized temperature-scaled cross entropy) Loss Function (Chen et al., 2020), which uses multiple negatives for a given (anchor, positive) pair, as using such a large number of negatives would significantly increase the compute (corresponding to the augmentation, forward pass, and backpropagation for a large number of inputs), which defeats **FastDoc**'s purpose.

### Hierarchical Classification using Hierarchical Labels.

Here we try to formulate a Supervised Hierarchical Classification Task based on a domain-specific hierarchical taxonomy. Given a document, the task is to predict the hierarchical categories present in the taxonomy.

In **FastDoc**, each document's representation is passed through $H$ classification heads, $H$ being the maximum number of hierarchical levels present in the taxonomy. It may so happen that the hierarchy for a document has less than $H$ levels. Hence, to bring uniformity, a 'null' class is added to each remaining level. For Hierarchical Classification, Local Classifier per Level (LCL) (Silla & Freitas, 2011) is used, where one multi-class classifier is trained for each level of hierarchy. At each level, a classification head is an MLP layer (followed by SoftMax). The hierarchical loss function $\mathcal{L}_{hier}$ is the sum of the categorical cross-entropy loss ($CELoss$) over all the $H$ classification heads, for all the $N$ input documents per training sample (in our case, $N = 3$). Mathematically,

$$\mathcal{L}_{hier} = \sum_{i=1}^{N} \sum_{j=1}^{H} CELoss(x_{ij}, y_{ij}), \tag{2}$$

$x_{ij}$ and $y_{ij}$ are predicted and target class distributions respectively, for the $i_{th}$ document, and $j_{th}$ classification head.

The loss $\mathcal{L}$ backpropagated during pre-training is the sum of the triplet margin loss and the hierarchical loss functions.

# 3 Pre-training Setup

We represent BERT-based and RoBERTa-based **FastDoc** as **FastDoc**$_{BERT}$ and **FastDoc**$_{RoBERTa}$ respectively, along with abbreviation of the domain (Customer Support - *Cus.*, Scientific Domain - *Sci.*, Legal Domain - *Leg.*). The proposed models and domain-specific baselines are pre-trained (in-domain) for 1 epoch. We use a batch size of 32, and AdamW optimizer (Loshchilov & Hutter, 2018) with an initial learning rate of $5 \times 10^{-5}$, which linearly decays to 0.

We next outline the specifics of the dataset used, its associated taxonomy, metadata leveraged. Table 1 shows examples of sample triplets and hierarchies from each domain.

| Domain, Data Source | Example Triplet | Example Hierarchy |
|---|---|---|
| Customer Support (E-Manuals Corpus) | stereo equalizer E-Manual, stereo equalizer E-Manual (of a different brand), *blu-ray player E-Manual* | **Stereo Equalizer** Electronics → Audio → Audio Players & Recorders → Stereo Systems |
| Scientific Domain (ArXiv) | Proximal Policy Optimization Algorithms Generating Natural Adversarial Examples *Autonomous Tracking of RF Source Using a UAV Swarm* | **Generating Natural Adversarial Examples** Computer Science → Machine Learning |
| Legal Domain (EURLEX57k) | "⋯ import licences ⋯ dairy products" "⋯ market research measures ⋯ milk and milk products" *"⋯ importations of fishery and aquaculture products"* | **"⋯ importation of olive oil ⋯"** Agriculture → Products subject to market organisation → Oils and fats |

Table 1: Examples of triplets and hierarchies in the 3 domains. (When representing triplets, we specify domain-specific metadata - product category in Customer Support, paper title in the Scientific Domain, and certain key phrases from each document in the Legal Domain) [2nd column] Underlined phrases denote "positive" documents; italicized phrases denote "negative" documents.

## 3.1 Pre-training in the Customer Support Domain

**Dataset and Triplets Chosen.** We pre-train **FastDoc** on a subset of the E-Manuals Corpus (Nandy et al., 2021) - we sample $2,000$ E-Manual triplets, such that, the anchor and positive E-Manuals belong to the same product category and the anchor and negative E-Manuals belong to different product categories. The amount of data is a mere 3% of the entire E-Manuals Corpus.

**Hierarchy considered.** Google Product Taxonomy (GPrT)[4] ($5,583$ possible hierarchies across 7 levels of hierarchy) is used to obtain hierarchical classification labels using (a single) category of an E-Manual. This allows similar E-manuals (e.g. 'TV' and 'Monitor') to have more similar hierarchies compared to dissimilar E-Manuals (e.g. 'TV' and 'Refrigerator'). Details on mapping product category to hierarchy are mentioned *in Section C.1 of Appendix.*

## 3.2 Pre-training in the Scientific Domain

**Dataset and Triplets Chosen.** We pre-train **FastDoc** on a subset of the ArXiv - we sample $2,000$ triplets of scientific papers based on the "primary category" assigned to the paper, such that, the anchor and positive papers belong to the same category, and the anchor and negative papers belong to different categories. For each such triplet, we add another triplet, where the positive and anchor samples are swapped. The amount of data used is negligible compared to the $1.14M$ Papers used by SciBERT (Beltagy et al., 2019) during its pre-training. Note that several recent works have used citations as a similarity signal (Cohan et al., 2020; Ostendorff et al., 2022; Yasunaga et al., 2022). However, a paper might cite another paper that is not similar in terms of the content. Instead, similarity based on "primary category" would more intuitively lead to content-based similarity.

**Hierarchy Considered.** ArXiv Category Taxonomy[5] (consisting of 155 possible hierarchies across 3 levels of hierarchy) is used to obtain hierarchical classification labels for each document, where each document is already mapped to its corresponding hierarchy via the taxonomy.

---

[4]https://support.google.com/merchants/answer/6324436?hl=en
[5]https://arxiv.org/category_taxonomy

### 3.3 Pre-training in the Legal Domain

**Triplets Chosen.** We pre-train *FastDoc* on a subset of the EURLEX57K dataset (Chalkidis et al., 2019) of legislative documents - we sample $2,000$ document triplets based on the list of annotated EUROVOC Concepts[6] assigned to each document, such that, the anchor and positive documents have at least 1 Concept in common, and the anchor and negative documents have no Concepts in common. We double the number of triplets in a way similar to Scientific Domain. The amount of data used is negligible compared to the 8GB of legal contracts used for domain-specific pre-training in Hendrycks et al. (2021).

**Hierarchy Considered.** The hierarchical class assignments of the documents in the EUR-Lex Dataset (Loza Mencia et al., 2010) (consisting of 343 possible hierarchies across 4 levels of hierarchy) are used as hierarchical classification labels, where each document is already mapped to its corresponding hierarchy.

## 4 Downstream Datasets/Tasks

The efficacy of the pre-training framework is tested through its performance in downstream tasks. We describe those tasks and the corresponding datasets used (The names of all tasks, their corresponding datasets, and domains are listed *in Table 15 of Section D of Appendix*).

### 4.1 Customer Support

We evaluate Question Answering Task on two datasets - single span QA on TechQA Dataset and multi-span QA on S10 QA Dataset (described *in Section D.1 of Appendix*).

**TechQA Dataset.** TechQA (Castelli et al., 2020) is a span-based QA dataset with questions from a technical discussion forum and the answers annotated using IBM Technotes, which are documents released to resolve specific issues. The dataset has 600 training, 310 dev, and 490 evaluation QA pairs. Each QA pair is provided with the document that contains the answer, along with 50 candidate Technotes retrieved using Elasticsearch[7].

**Fine-tuning Setup.** The fine-tuning is carried out in two stages - first on the SQuAD 2.0 Dataset (inspired by Castelli et al. (2020)), and then on task-specific QA datasets, which is discussed *in Section E.1 of Appendix*). Note that results without intermediate fine-tuning on SQuAD 2.0 deteriorate, as shown *in Section E.1 of Appendix*.

### 4.2 Scientific Domain

We use multiple datasets from **SciBERT Benchmark Datasets** (mentioned in Beltagy et al. (2019)) for training and evaluation. The following downstream tasks and corresponding datasets are used for evaluation - (1) NER (Named Entity Recognition): We use the **BC5CDR** (Li et al., 2016), **JNLPBA** (Kim et al., 2004), and **NCBI-Disease** (Doğan et al., 2014) NER Datasets of the Biomedical Domain. (2) REL (Relation Classification): This task predicts the type of relation between entities. The **ChemProt Dataset** (Kringelum et al., 2016) from the Biomedical Domain and **SciERC Dataset** (Luan et al., 2018) from the Computer Science Domain are used for evaluation. (3) CLS (Text Classification): **SciCite Dataset** (Cohan et al., 2019) gathered from Multiple Domains is used.

**Fine-tuning Setup.** We fine-tune and evaluate on the downstream tasks mentioned above. The hyperparameters are the same as that in Beltagy et al. (2019).

### 4.3 Legal Domain

**CUAD** (Contract Understanding Atticus Dataset) (Hendrycks et al., 2021) is used, which is annotated by legal experts for the task of Legal Contract Review. It consists of $13,101$ clauses across 41 types of clauses

---

[6]http://eurovoc.europa.eu/
[7]https://www.elastic.co/products/elasticsearch

annotated from 510 contracts. Given a contract, for each type of clause, the task requires extracting relevant clauses as spans of text related to the clause type. Details of the dataset splits are given *in Section D.3 of Appendix.*

**Fine-tuning Setup.** We fine-tune and evaluate on the Contract Review Task on CUAD. The hyperparameters are the same as that in Hendrycks et al. (2021).

## 5  Experiments and Results

To assess the performance of our proposed methods, we fine-tune and evaluate these methods and baselines on the datasets described in Section 4, and draw inferences. Due to space constraints, the performance on the S10 QA Dataset is reported *in Section E.1 of Appendix*. In all these experiments, we perform an ablation study by considering each of the two losses of *FastDoc* separately i.e. we use only Triplet Loss (*triplet*) and only Hierarchical Classification Loss (*hier.*). We perform several additional ablations (see *Section E of Appendix*) - (1) Pre-training both lower and higher-level encoders (entire hierarchical architecture), followed by fine-tuning the lower encoder worsens performance, suggesting - higher-level encoder learns better task-specific features (2) replacing sBERT/sRoBERTa with BERT/RoBERTa worsens performance, suggesting - sentence transformers provide effective sentence embeddings. (3). used a more fine-grained document similarity criterion (changed Eq. 1) and found the result to be inferior, and (4). compared *FastDoc* with the much larger GPT-3.5 model and found that GPT-3.5 models perform much inferior in 0 and 1-shot settings.

### 5.1  Customer Support Domain

**Baselines:** We compare our pre-training approach to 3 types of pre-training baselines described below. For the sake of completeness, we also compare with baselines using span/sentence-level supervision signals. Domain-specific Continual Pre-training is carried out on the corpus of E-Manuals for all baselines (except $BERT_{BASE}$, $RoBERTa_{BASE}$, and Longformer).
**(1) Pre-training using masked language modeling (MLM) and/or Next Sentence Prediction (NSP)**: We use $BERT_{BASE}$ (Devlin et al., 2019), $RoBERTa_{BASE}$ (Liu et al., 2019), $LinkBERT_{BASE}$ Yasunaga et al. (2022), Longformer (Beltagy et al., 2020), $EManuals_{BERT}$ and $EManuals_{RoBERTa}$ (Nandy et al., 2021) (domain continual pre-training of $BERT_{BASE}$ (Devlin et al., 2019) and $RoBERTa_{BASE}$ (Liu et al., 2019), respectively, on the entire E-Manuals corpus). **(2) Using intra-document contrastive learning**: DeCLUTR (Giorgi et al., 2021) and ConSERT (Yan et al., 2021) are the intra-document contrastive learning methods. **(3) Using inter-document contrastive learning**: SPECTER (Cohan et al., 2020) is the inter-document contrastive learning baseline used. (Details on tailoring SPECTER to Customer Support are given *in Section E.1 of Appendix*).

**Performance on TechQA Dataset** The answer-retrieval performance on the development set (as per Castelli et al. (2020)) is reported in Table 2. The model gives five candidate answers per question and corresponding confidence scores. Each answer is assigned an 'evaluation score' - If the confidence score is below a threshold provided by the model, 'evaluation score' is 1 if the question is actually unanswerable, and 0 otherwise. However, if the confidence score is above the threshold, the 'evaluation score' is character F1 between the predicted answer and ground truth and 0 if the question is actually unanswerable. The evaluation metrics used, as mentioned in Castelli et al. (2020)[8], are (a). **F1** - 'evaluation score' for the predicted answer (with the highest confidence score) averaged across all questions. (b). **HA_F1@1** - similar to F1, except that, the averaging is done on the answerable question set (160 out of 310 questions in the dev set are answerable). (c). **HA_F1@5** - macro average of the 5 best candidate answers per question, averaged across the answerable question set.

From the results in Table 2, we can infer - (1). Among the baselines, (a) Longformer gives the best F1 and HA_F1@1 and the second-best HA_F1@5. This is because of the long sequence length of $4,096$ compared to

---

[8]We do not use BEST_F1, as a threshold is tuned on the dev. set using F1 score, which is not realistic

| | F1 | HA_F1@1 | HA_F1@5 |
|---|---|---|---|
| $BERT_{BASE}$ | 13.67 | 26.49 | 36.14 |
| $RoBERTa_{BASE}$ | 16.46 | 31.89 | 42.4 |
| $LinkBERT_{BASE}$ | 14.24 | 27.59 | 36.77 |
| Longformer | 16.57 | 32.1 | 42.66 |
| $EManuals_{BERT}$ | 13.41 | 25.98 | 36.69 |
| $EManuals_{RoBERTa}$ | 16.04 | 31.08 | 44.71 |
| DeCLUTR | 15.11 | 29.28 | 38.93 |
| ConSERT | 11.12 | 21.54 | 30.37 |
| SPECTER | 12.92 | 25.03 | 34.74 |
| ***FastDoc(Cus.)***$_{BERT}$*(hier.)* | 14.19 | 27.49 | 36.62 |
| ***FastDoc(Cus.)***$_{BERT}$*(triplet)* | 14.47 | 28.04 | 37.21 |
| ***FastDoc(Cus.)***$_{BERT}$ | 14.56 | 28.2 | 35.54 |
| ***FastDoc(Cus.)***$_{RoBERTa}$*(hier.)* | 16.52 | 32.00 | 44.77 |
| ***FastDoc(Cus.)***$_{RoBERTa}$*(triplet)* | 16.39 | 31.76 | **46.59** |
| ***FastDoc(Cus.)***$_{RoBERTa}$ | **17.52** | **33.94** | 44.96 |

Table 2: Results for the QA task on the TechQA Dataset.

512 of other models[9]. (b) SPECTER does not perform well, even though it uses document-level supervision, as it cannot accommodate the entire document within 512 tokens, so only the first 512 tokens are used which does not help much in learning. (c) ConSERT performs contrastive learning on sentence inputs, prohibiting it from learning context beyond a single sentence (unlike ***FastDoc*** that learns inter-sentence context during pre-training due to its hierarchical architecture), thus reducing performance on QA tasks. (d) In general, contrastive learning baselines perform inferior to those using MLM/NSP. (2). ***FastDoc(Cus.)***$_{BERT}$ variants perform better than BERT-based baselines, and ***FastDoc(Cus.)***$_{RoBERTa}$ variants than almost all RoBERTa-based baselines, suggesting that our proposed pre-training methods are better than that of baselines. (3) ***FastDoc(Cus.)***$_{RoBERTa}$ variants perform better than ***FastDoc(Cus.)***$_{BERT}$ variants, as RoBERTa (Liu et al., 2019) performs better than BERT (Devlin et al., 2019) in span-based QA tasks such as SQuAD (Rajpurkar et al., 2018; 2016). (4) ***FastDoc(Cus.)***$_{RoBERTa}$ performs the best of all models in F1 and HA_F1@1 and the second-best in HA_F1@5. ***FastDoc(Cus.)***$_{RoBERTa}$ performs around 6% better than the best baseline Longformer both in terms of F1 and HA_F1@1 (even though Longformer has a long sequence length, it is not able to encode most documents properly).

Additionally, we perform a qualitative analysis ***in Table 20 of Section E.1 of Appendix*** by comparing the ground-truth answers and the answers predicted by ***FastDoc*** and a well-performing baseline for 3 answerable questions in TechQA and S10 QA Datasets. This analysis suggests that ***FastDoc*** is comparatively better at extracting numerical entities, tackling multiple questions in a sample, and answering location-based questions.

### 5.2 Scientific Domain

**Baselines:** SciBERT (Beltagy et al., 2019) (pre-trained using MLM and NSP on a huge scientific corpus)[10].

**Performance on Different Datasets** We fine-tune and evaluate on the datasets mentioned in Section 4.2. The results on the test set for each task are shown in Table 3. We see that ***FastDoc***$(Sci.)_{BERT}$ performs better than SciBERT on 4 out of 6 datasets, and performs the best on the Relation Classification Tasks. However, ***FastDoc***$(Sci.)_{BERT}$*(hier.)* performs the best on 3 datasets with NER and text classification tasks, as (1) fine-grained NER benefits from fine-grained hierarchical information, and (2) text classification dataset has samples from multiple domains, where diversity in the hierarchical categories helps.

---

[9]For completeness, we have also continually pre-trained Longformer on the data used by ***FastDoc***, and it shows inferior results to Longformer on 2/3 metrics due to the data being insufficient to adapt Longformer.

[10]vocabulary used for SciBERT is same as that of $BERT_{BASE}$ for consistency among SciBERT and ***FastDoc*** variants. Specifically, we use this model as SciBERT - `https://s3-us-west-2.amazonaws.com/ai2-s2-research/scibert/huggingface_pytorch/scibert_basevocab_uncased.tar`

| Field | Task | Dataset | SciBERT | *FastDoc (triplet)* | *FastDoc (hier.)* | *FastDoc* |
|-------|------|---------|---------|---------------------|-------------------|-----------|
| BIO | NER | BC5CDR | 85.55 | 87.7 | **87.94** | 87.81 |
| | | JNLPBA | 59.5 | 75.86 | **75.97** | 75.84 |
| | | NCBI-D | **91.03** | 84.15 | 87.81 | 84.33 |
| | REL | ChemProt | 78.55 | 75.12 | 80.28 | **80.48** |
| CS | REL | SciERC | 74.3 | 75.4 | 75.62 | **78.95** |
| Multi | CLS | SciCite | 84.44 | 84.31 | **84.48** | 83.59 |

Table 3: **FastDoc**$(Sci.)_{BERT}$ and its variants vs. SciBERT in tasks presented in Beltagy et al. (2019). Following Beltagy et al. (2019), we report macro F1 for NER (span-level), and for REL and CLS (sentence-level), except for ChemProt, where we report micro F1.

Since recent works have used citations as a similarity signal, we report the performance of **FastDoc** using citations as a similarity signal *in Table 22 of Section E.2 of the Appendix.* This gives a satisfactory performance, showing that **FastDoc** works on different types of metadata. However, on average, a system using citations does not perform as well as when using "primary category".

### 5.3 Legal Domain

**Baselines:** We use baselines from Hendrycks et al. (2021) - BERT$_{BASE}$ , RoBERTa$_{BASE}$ , RoBERTa$_{BASE}$ + Contracts Pre-training (domain-specific pre-training of RoBERTa-BASE on ~8GB of unlabeled contracts collected from the EDGAR database). Also, we use CDLM Caciularu et al. (2021), LEGAL-BERT-FP (Chalkidis et al., 2020), and LEGAL-RoBERTa-BASE (Geng et al., 2021) as additional baselines.

**Performance on CUAD Dataset**

| Model | AUPR | Precision@ 80% Recall) |
|-------|------|------------------------|
| BERT$_{BASE}$ | 32.4 | 8.2 |
| LEGAL-BERT-FP | 32.6 | 21.16 |
| RoBERTa$_{BASE}$ | 42.6 | 31.1 |
| LEGAL-RoBERTa-BASE | 42.9 | 31.7 |
| RoBERTa$_{BASE}$ + Contracts Pre-training | **45.2** | 34.1 |
| CDLM | 43.2 | **34.6** |
| **FastDoc**$(Leg.)_{BERT}(triplet)$ | 32.5 | 8.3 |
| **FastDoc**$(Leg.)_{BERT}(hier.)$ | 32.8 | 9.4 |
| **FastDoc**$(Leg.)_{BERT}$ | 32.6 | 9.4 |
| **FastDoc**$(Leg.)_{RoBERTa}(triplet)$ | 42.4 | 32.7 |
| **FastDoc**$(Leg.)_{RoBERTa}(hier.)$ | 42 | 32.3 |
| **FastDoc**$(Leg.)_{RoBERTa}$ | 44.8 | **34.6** |

Table 4: **FastDoc**$(Leg.)$ and its variants vs. baselines in the Contract Review task on CUAD (AUPR - Area Under Precision-Recall Curve).

CUAD exhibits class imbalance, rendering AUPR (Area Under Precision-Recall) and Precision@80% Recall as suitable metrics. Furthermore, AUPR effectively encapsulates model performance across various confidence thresholds. From Table 4, we infer - (1) All **FastDoc**$(Leg.)_{BERT}$ variants perform better than BERT$_{BASE}$, and **FastDoc**$(Leg.)_{RoBERTa}$ performs better than RoBERTa$_{BASE}$ and Legal-RoBERTa-BASE (2) **FastDoc**$(Leg.)_{RoBERTa}$ performs better than CDLM, even though CDLM has a long sequence length of 4096 (3) **FastDoc**$(Leg.)_{RoBERTa}$ gives the best Precision@80% Recall, and second-best AUPR, even though it uses negligible domain-specific pre-training data compared to baselines.

**Summary of the Experiments and Results.** Note that although **FastDoc** uses document information, it does not use the information derived from MLM during continual pre-training that many other conventional domain-specific baselines use. Therefore, we maintain that our approach is both equitable and innovative

when compared to the baselines. We observe that across various domains and different types of generalization, **FastDoc** typically outperforms (albeit modestly) the baselines, and in cases where it falls short, the difference is marginal. This happens despite using much less compute which is elaborated in Section 6.

### 5.4 Utility of the pre-training losses: Examples

We looked into the datasets and gauged the impact of the two pre-training losses in **FastDoc**'s performance. Specifically, we took cases where **FastDoc** has produced better results than a well-performing baseline and chose some representative examples to present. Table 5 shows examples from each domain where triplet and hierarchical losses are beneficial, along with probable reasons. We can see that domain-specific knowledge present in the metadata and taxonomy helps **FastDoc** in performing well on domain-specific downstream tasks.

| | Dataset | **Triplet Loss is beneficial** | **Hierarchical Loss is beneficial** |
|---|---|---|---|
| **Cus.** | S10 QA (QA) | Q. How can I enable the accidental touch protection ? | Q. I need the registered fingerprint list. Where can I find this? |
| | Reasons | "accidental touch" benefits from triplets having anchor and positive as Touch-based device E-Manuals, and negative as an E-Manual of a device without touch screen. | "fingerprint" benefits from multiple hierarchies with "Biometric Monitors". |
| **Sci.** | SciCite (Text Classification) | A primary benefit of these models is the inclusion of variability in model parameters (Parnell et al. 2010). **Output - "background"** | The SVR can be considered as a novel training technique; the following section presents a concise introduction to the SVR [33, 35, 38]. **Output - "background"** |
| | Reasons | Using triplets with anchor and positive belonging to "Machine Learning" help in classifying the text as "background", as "model parameters" is a common term used in "Machine Learning" papers. | Although the first sentence could lead to the inference that "SVR" is a new method, other papers belonging to the hierarchy "Computer Science → Machine Learning" would suggest that "SVR" exists already. |
| **Leg.** | CUAD (Clause Extraction) | ..to make or have made the Products anywhere in the world for **import** or **sale** in the Field in the Territory in each case,.. | ..such **commercial crops** will be interplanted as **agriculture** and forestry as well as medicinal materials;.. |
| | Reasons | The triplet on the concepts of "import policy" and "sale" in the anchor and positive is beneficial. | the hierarchy "Agriculture → Products subject to market organisation" helps here. |

Table 5: Samples from each domain, where Triplet Loss and Hierarchical Loss are beneficial. Note that we add outputs for the SciCite Text Classification Task for more clarity.

## 6 Pre-training Compute of *FastDoc* relative to the baselines

| Domain | Model | Compute (in GPU-hours) |
|---|---|---|
| Customer Support | EManuals$_{BERT}$ | 576 |
| | EManuals$_{RoBERTa}$ | 980 |
| | DeCLUTR | 370 |
| | ConSERT | 40 |
| | SPECTER | 600 |
| | **FastDoc**$(Cus.)_{BERT}$ | 0.58 |
| | **FastDoc**$(Cus.)_{RoBERTa}$ | 0.75 |
| Scientific Domain | SciBERT | 7680 |
| | **FastDoc**$(Sci.)_{BERT}$ | 1.7 |
| Legal Domain | RoBERTa$_{BASE}$ + Contracts Pre-training | 710 |
| | **FastDoc**$(Leg.)_{RoBERTa}$ | 1.49 |

Table 6: Pre-training Compute of **FastDoc** vs. baselines

We compare the compute (in terms of GPU-hours - GPUs needed multiplied by the number of hours) for pre-training ***FastDoc*** with baselines. NVIDIA GeForce GTX 1080 Ti GPUs are used for pre-training.

Baselines in Customer Support and Legal Domains are continually pre-trained on the domain, while the SCIBERT baseline in Scientific Domain is pre-trained from scratch.

**Customer Support:** Table 6 shows that ***FastDoc(Cus.)$_{BERT}$*** and ***FastDoc(Cus.)$_{RoBERTa}$*** use roughly 1,000 times[11] and 1,300 times less compute compared to EManuals$_{BERT}$ and EManuals$_{RoBERTa}$ respectively, and require significantly less compute than all the baselines. *It actually takes less than 1 GPU-hour for pre-training **FastDoc**.* ConSERT is the closest baseline in terms of compute time, as its inputs are a limited number of sentence pairs, unlike a huge number of spans in DeCLUTR, a large number of triplets in SPECTER, and several masked sentences in EManuals$_{BERT}$ and EManuals$_{RoBERTa}$ . **Legal Domain:** ***FastDoc***(*Leg.*)$_{RoBERTa}$ needs around 480 times less compute than continual pre-training of RoBERTa-BASE on contracts as in Hendrycks et al. (2021).

**Scientific Domain:** ***FastDoc***(*Sci.*)$_{BERT}$ needs around $4,520$ times less compute than SCIBERT[12]. The decrease in compute of ***FastDoc*** compared to the domain-specific baselines is much more compared to Customer Support and Legal Domains, as SCIBERT is pre-trained from scratch. ***FastDoc***(*Sci.*)$_{BERT}$ continually pre-trains BERT$_{BASE}$ on Scientific Domain, and its downstream task performance and domain-specific compute compared to SCIBERT shows that *domain-specific pre-training from scratch is not necessary.*

Thus, these experiments demonstrate the remarkable efficiency of the proposed pre-training paradigm, as well as the choice of the pre-training architecture used in ***FastDoc***.

## 7 Analysis and Ablations

In this section, we report the following analysis and ablations - (1) Catastrophic Forgetting when evaluating ***FastDoc*** in open-domain (2) absence of document supervision in the domain of interest (3) Effect of using a larger backbone model for ***FastDoc*** (4) Reasons behind ***FastDoc*** working the way it does. Also, we apply Parameter-Efficient training on ***FastDoc*** as an ablation ***in Section G.2 of Appendix***, which gives very poor downstream task results and is not beneficial from a compute perspective as well.

| TASK | CoLA | SST2 | MRPC | | STS | | QQP | | MNLI | QNLI | RTE |
|---|---|---|---|---|---|---|---|---|---|---|---|
| **METRIC** | **Matthews CC** | **Acc.** | **F1-score** | **Acc.** | **Pearson CC** | **Spearman CC** | **F1-score** | **Acc.** | **Acc.** | **Acc.** | **Acc.** |
| RoBERTa$_{BASE}$ | **63.71** | 94.15 | 92.71 | 89.71 | 90.91 | **90.66** | **89.1** | **91.84** | **87.24** | 92.26 | **80.14** |
| ***FastDoc(Cus.)$_{RoBERTa}$*** | 62.57 | **94.27** | **93.1** | **90.44** | **90.98** | **90.66** | 89.08 | 91.84 | 87.22 | **92.62** | 79.06 |
| | (-1.14) | (+0.12) | (+0.39) | (+0.73) | (+0.07) | (0) | (-0.02) | (0) | (-0.02) | (+0.36) | (-1.08) |
| EManuals$_{RoBERTa}$ | 51.82 | 91.97 | 91.42 | 87.99 | 88.4 | 88.36 | 88.65 | 91.55 | 85.15 | 91.34 | 70.4 |
| | (-11.89) | (-2.18) | (-1.29) | (-1.72) | (-2.51) | (-2.3) | (-0.45) | (-0.29) | (-2.09) | (-0.92) | (-9.74) |

Table 7: Dev. set results on GLUE Benchmark (CC - Correlation Co-efficient, Acc. - Accuracy)

### 7.1 Catastrophic Forgetting in open-domain

Recent works show that continual in-domain pre-training of transformers leads to a significant performance drop when fine-tuned on open-domain datasets (Arumae et al., 2020; Gururangan et al., 2020) resulting in Catastrophic Forgetting (CF). Such works start with an open-domain model (e.g. BERT/RoBERTa) and perform open-domain benchmark tasks (e.g. GLUE). Then, they consider a model pre-trained continually on a specific domain (e.g. BioBERT) and re-assess performance on those tasks. Decrease in performance of domain-specific model compared to the open-domain model determines degree of catastrophic forgetting.

Similarly, we fine-tune RoBERTa$_{BASE}$ (pre-trained on open-domain corpora), ***FastDoc(Cus.)$_{RoBERTa}$***, and EManuals$_{RoBERTa}$ from **customer support** on the datasets of the (open-domain) GLUE (Wang et al., 2018)

---

[11]***FastDoc(Cus.)$_{BERT}$*** uses 33.3x less documents compared to EManuals$_{BERT}$ during pre-training. Additionally, ***FastDoc(Cus.)$_{BERT}$*** takes sentence embeddings as inputs, while EManuals$_{BERT}$ takes in token embeddings. There are 37.3 tokens per sentence in the pre-training corpus, meaning that there are 37.3x lesser samples for ***FastDoc(Cus.)$_{BERT}$*** w.r.t EManuals$_{BERT}$ for the same text, reducing compute further from 33.3x to 1000x ($33.3 \times 37$ is $\approx 1000$)

[12]According to Beltagy et al. (2019), it takes a minimum of 40 days on 8 GPUs (elaborated ***in Section F of Appendix***)

Figure 2: Relative change (in $Log_{10}$ Scale) in the L1-norm of different types of parameters during pre-training via MLM vs. *FastDoc*.

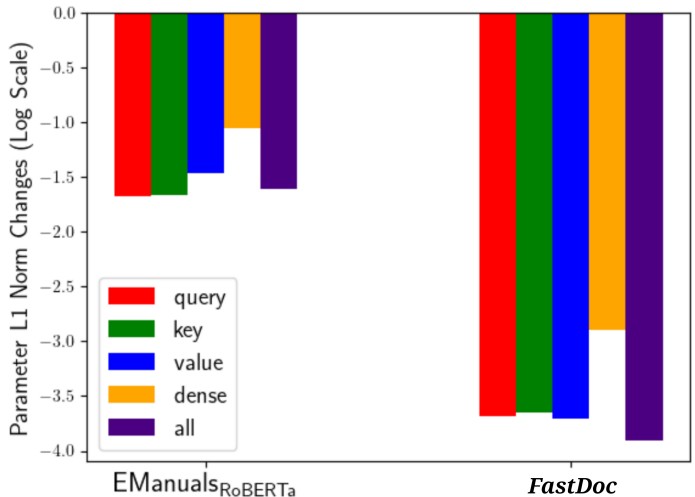

benchmark, and the results are shown in Table 7. The hyperparameters used are mentioned ***in Section G.1 of Appendix.***

We observe that - (a). ***FastDoc(Cus.)$_{RoBERTa}$*** performs better than RoBERTa$_{BASE}$ in 4 out of 8 tasks (although the improvement is minor), even after continual pre-training on E-Manuals, while the drop in performance in the other 4 tasks is negligible. The performance improvement in tasks that require predicting relation between sentence pairs, like STS, QNLI, MRPC, could be attributed to the Contrastive Learning Objective when pre-training ***FastDoc*** (b). EManuals$_{RoBERTa}$ performs considerably worse compared to RoBERTa$_{BASE}$ on all tasks, suggesting that MLM is not robust against domain change. The possible reason behind the superior performance of ***FastDoc(Cus.)$_{RoBERTa}$*** is that it **requires only a small fraction of pre-training data** compared to what is used by domain-specific baselines such as EManuals$_{RoBERTa}$, hence making only **small changes in the parameter space** that helps retain open-domain knowledge while learning essential domain-specific knowledge. We perform an experiment to test the proposition and plot the relative change in L1-norm of different types of parameters such as attention query, key, value matrices, and dense MLP parameters (similar to Wu et al. (2022)) during pre-training via MLM vs. ***FastDoc***, as shown in Figure 2. We observe that the relative change of parameters in ***FastDoc*** is about 100 times less compared to MLM.

## 7.2 Absence of document level information

| Model | F1 | HA__F1@1 | HA__F1@5 |
|:---:|:---:|:---:|:---:|
| RoBERTa$_{BASE}$ | 16.46 | 31.89 | 42.4 |
| ***FastDoc***$(Cus.)_{RoBERTa}$ (7 hier. levels) | 17.52 (+6.44%) | 33.94 (+6.43%) | 44.96 (+6.04%) |
| ***FastDoc***$(Cus.)_{RoBERTa}$ (w/o est. meta., tax., 7 hier. levels) | 15.39 (-6.5%) | 29.83 (-6.46%) | 44.35 (+4.6%) |
| ***FastDoc***$(Cus.)_{RoBERTa}$ (w/o est. meta., tax., 15 hier. levels) | **18.01 (+9.42%)** | **34.89 (+9.41%)** | **47.53 (+12.1%)** |

Table 8: Results on TechQA Dataset in Customer Support Domain with and without established domain-specific document metadata and taxonomy

A pre-requisite of ***FastDoc*** has been the availability of document metadata and taxonomy. In this experiment, we go beyond that and derive document similarity via similarity based on the ROUGE-L score among

documents, followed by creating a custom taxonomy of document category hierarchies using Hierarchical Topic Modeling (Grootendorst, 2022). Table 8 shows results on the Customer Support (see other domains' results *in Section G.3 of Appendix*), and we can see that gives comparable performance when considering same number of hierarchical levels as *FastDoc*. However, since the taxonomy is derived using topic modeling, we are here not constrained by the number of hierarchies. We notice that the performance improves when a larger number of hierarchical levels are used, showing great potential for adapting *FastDoc* to any domain of interest. However, note that even though one can devise a (unsupervised) way to extract triplets and document hierarchies, it is much more efficient to use metadata and taxonomy if and when available, as there is some time and CPU involved in deriving content-similarity-based metrics like ROUGE-L score due to the large size of the documents.

### 7.3 Effect of using a larger backbone model for *FastDoc*

| Model | F1 | HA_F1@1 | HA_F1@5 |
|---|---|---|---|
| $FastDoc(Cus.)_{RoBERTa}$ | 17.52 | 33.94 | 44.96 |
| $FastDoc(Cus.)_{RL}$ | **18.48** (+5.48%) | **35.8** (+5.48%) | **47.8** (+6.32%) |

Table 9: Results on TechQA Dataset in Customer Support Domain (RL - RoBERTa-LARGE)

| Field | Task | Dataset | $FastDoc(Sci.)_{BERT}$ | $FastDoc(Sci.)_{BL}$ |
|---|---|---|---|---|
| BIO | NER | BC5CDR | 87.81 | **88.45** (+0.73%) |
| | | JNLPBA | 75.84 | **76.53** (+0.91%) |
| | | NCBI-D | 84.33 | **86.18** (+2.19%) |
| | REL | ChemProt | 80.48 | **84** (+4.37%) |
| CS | REL | SciERC | 78.95 | **80.26** (+1.66%) |
| Multi | CLS | SciCite | 83.59 | **85.76** (+2.6%) |

Table 10: Results on tasks presented in Beltagy et al. (2019) (BL - BERT-LARGE)

| Model | AUPR | Precision@ 80% Recall |
|---|---|---|
| $FastDoc(Leg.)_{RoBERTa}$ | 44.8 | 34.6 |
| $FastDoc(Leg.)_{RL}$ | **45.3** (+1.12%) | **39.5** (+14.16%) |

Table 11: Results on CUAD Dataset in Legal Domain (RL - RoBERTa-LARGE)

Tables 9, 10, and 11 compare the impact of using a larger backbone compared to the one used in the proposed *FastDoc* (e.g. RoBERTa-LARGE vs. RoBERTa-BASE, BERT-LARGE vs. BERT-BASE). From the results, we can see that using a larger model as a backbone further improves results due to an increased number of trainable parameters.

### 7.4 Analysis of the interoperability of embeddings

*FastDoc* shows that using input sentence embeddings during pre-training helps when using token embedding inputs during fine-tuning, as is evident from the potent downstream task performance. We analyze this **interoperability of embeddings** by answering the following research questions (observations and experiments elaborated *in Section G.4 of Appendix*) - (a). *How does FastDoc learn local context?* - Similar documents have very-similar local (paragraph-level) contexts, suggesting that, using document-level supervision during pre-training implicitly learns local context. Also, in an experiment, we randomly sample 500 sentences from each of the 3 domains. For each sentence, we mask a random token and calculate the change in its prediction probability on masking other tokens in the sentence. Spearman Correlation of this change between *FastDoc* and a domain-specific model pre-trained using MLM is moderately high for all domains, showing that **local context is learned by *FastDoc* to a reasonable extent**. (b). *Are relative*

*representations preserved across the two embedding spaces?* - Independent of whether inputs are sentence or token embeddings, documents are clustered in a similar manner across the two representation spaces, hence, **relative representations are preserved**.

## 8 Prior Art

**Representation Learning using self-supervised learning methods:** In recent times, downstream tasks in NLP use representation learning techniques where transformers are pre-trained on large text corpora using self-supervised learning methods like NSP (Devlin et al., 2019), MLM (Devlin et al., 2019; Liu et al., 2019), contrastive learning (Giorgi et al., 2021; Yan et al., 2021; Wang et al., 2021; Cohan et al., 2020), etc. before fine-tuning on downstream tasks. There are models pre-trained on domain-specific corpora such as E-Manuals (Nandy et al., 2021), legal texts (Chalkidis et al., 2020), bio-medical documents (Lee et al., 2020), etc.

**Supervised Pre-training:** Feng et al. (2022) proposes supervised pre-training on Leave-One-Out KNN that improves transfer to downstream tasks. CLMSM (Nandy et al., 2023) uses recipe metadata as supervision signal for pre-training. MVP (Tang et al., 2023) leverages labeled data from a corpus across 11 tasks for pre-training, by unifying the data into text-to-text format. The paper also states that - unsupervised pre-training likely incorporates noise that affects the downstream performance, making supervised pre-training a better alternative. CLIP (Radford et al., 2021) utilizes the pre-training task of predicting which caption goes with which image (natural language supervision), which is an efficient way to learn image representations.

**Incorporating hierarchical information for enhancing representations:** Hierarchical information in the form of taxonomy and ontology has been used by some works to enhance learned representations. Barkan et al. (2021) introduces a Variational Bayes entity representation model that leverages additional hierarchical and relational information. Barkan et al. (2020) also uses a similar Bayesian approach to produce better representations, especially for rare words.

**Intra-document Contrastive Learning:** DeCLUTR (Giorgi et al., 2021) uses a DistilRoBERTa-base (Sanh et al., 2019) encoder. Spans overlapping or subsuming each other are considered as similar inputs, and other spans are considered as dissimilar inputs. InfoNCE Loss Function (Sohn, 2016) brings representations of similar spans closer and pushes representations of dissimilar spans farther away. ConSERT (Yan et al., 2021) also uses contrastive loss, but it performs sentence augmentation using adversarial attack (Kurakin et al., 2016), token shuffling, etc. It considers a sentence and its augmented counterpart to be similar, and any other sentence pair as dissimilar. CLINE (Wang et al., 2021) creates similar and dissimilar samples from a sentence by replacing some word(s) with their synonyms and antonyms using WordNet (Miller, 1995) and then uses contrastive loss.

**Inter-document contrastive learning:** SPECTER (Cohan et al., 2020) uses a triplet margin loss to pull similar documents closer to each other, and dissimilar ones are pushed away. The document representations are obtained using a transformer encoder. However, their encoder is only able to encode a maximum of 512 tokens of a document. SDR (Ginzburg et al., 2021) uses a self-supervised method by combining MLM loss and Contrastive Loss to learn document similarity. LinkBERT (Yasunaga et al., 2022) adds a Document Relation Prediction Objective to MLM during pre-training, where the task is to predict whether two segments are contiguous, random, or from linked documents. CDLM (Caciularu et al., 2021) leverages document-level supervision by applying MLM over a set of related documents using Longformer (Beltagy et al., 2020). These works are in line with our work, but they are unable to tackle the important technical challenges of large input size and scalability and in turn, suffer from the problems of limited input size and high pre-training compute.

## 9 Summary and Conclusion

Recent studies have repeatedly stressed the importance of domain-specific pretraining but also pointed to the costly and elaborate operation that must be undertaken to achieve reasonable performance. This paper shows that leveraging 1) document-level semantics, and 2) interoperability of input sentence embeddings (during

pre-training) and token embeddings (during fine-tuning), substantially reduces the compute requirements for domain-specific pre-training by at least 500 times, even while achieving better results on 6 different downstream tasks and 9 different datasets. The frugal pretraining technique has an important side-effect, it shows negligible *catastrophic forgetting* on the open-domain GLUE Benchmark. We also demonstrate that the existence of well-defined metadata and taxonomy is not mandatory; ***FastDoc*** performs effectively when discovering such metadata and taxonomy through unsupervised methods, illustrating its potential for future application across various domains.

**Limitations**

- ***FastDoc*** is robust to a wide document similarity range across several domains. However, performance in presence of high levels of noise in metadata is not guaranteed and further investigation is required to characterize it.

- Applicability of the proposed model to decoder-only and encoder-decoder models: ***FastDoc*** can be extended to decoder-only models like GPT-2 Radford et al. (2019), and encoder-decoder models like BART-BASE Lewis et al. (2020). We apply ***FastDoc*** using GPT-2 backbone (referred to as ***FastDoc***$_{GPT-2}$) and the BART-BASE encoder as backbone (referred to as ***FastDoc***$_{BART-BASE}$). Downstream task is dialogue summarization (i.e., a text generation task) on TweetSumm Dataset Feigenblat et al. (2021) in the Customer Support Domain. We compare it with GPT-2 and BART-BASE.

| Model | ROUGE-1 | ROUGE-2 | ROUGE-L |
|---|---|---|---|
| GPT-2 | 0.151 | 0.066 | 0.119 |
| ***FastDoc***$_{GPT-2}$ | 0.134 | 0.058 | 0.104 |

Table 12: Results of ***FastDoc***$_{GPT-2}$ vs. GPT-2 on TweetSumm Dataset

| Model | ROUGE-1 | ROUGE-2 | ROUGE-L |
|---|---|---|---|
| BART-BASE | 0.523 | 0.314 | 0.472 |
| ***FastDoc***$_{BART-BASE}$ | 0.524 | 0.315 | 0.473 |

Table 13: Results of ***FastDoc***$_{BART-BASE}$ vs. BART-BASE on TweetSumm Dataset

Tables 12 and 13 show that ***FastDoc*** gives poor results when using a decoder-only model as the backbone, and gives negligible improvement when using the encoder of an encoder-decoder model as the backbone. Improvement in results needs changing the architecture and document supervision objectives used in ***FastDoc*** to adapt to decoder and encoder-decoder models end-to-end. One way to adapt ***FastDoc*** to decoder model backbones is hierarchical decoding (like hierarchical encoding in ***FastDoc***) in 2 stages - decoding special, representative sentence tokens, which are then used to decode subword tokens. This is a potential future work.

- Applicability of the proposed model when downstream tasks are generation tasks: Tables 12 and 13 show that ***FastDoc***$_{GPT-2}$ and ***FastDoc***$_{BART-BASE}$ do not perform well on a text generation task. Improvement in results could be attained in a manner mentioned above.

**Broader Impact Statement**

The proposed methodology is, in general, applicable to any domain. Specifically, it can potentially be applied to user-generated text available on the web and is likely to learn patterns associated with exposure bias. This needs to be taken into consideration before applying this model to user-generated text crawled from the web. Further, like many other pre-trained language models, interpretability associated with the output is rather limited, hence users should use the output carefully.

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

## Appendix

The Appendix is organized in the same sectional format as the main paper. The additional material of a section is put in the corresponding section of the Appendix so that it becomes easier for the reader to find the relevant information. Some sections and subsections may not have supplementary material so only their name is mentioned. The page numbers are in continuation from the Main Paper's end page number.

## A Introduction

## B *FastDoc* Framework

Figure 3 shows the percentage of documents encoded entirely by RoBERTa-BASE encoder when the input is 512 tokens vs. 512 sentences.

Figure 4 shows a detailed overview of the **FastDoc** Framework.

## C Pre-training Setup

### C.1 Pre-training in the Customer Support Domain

Table 14 shows 4 examples of E-manual product categories and the hierarchies assigned to them with the help of the GPrT. We can see that more similar products tend to have more similar hierarchies.

**Mapping E-Manual product category to hierarchy:** It may so happen that the product category of the E-Manual does not have an exact match with any leaf category in the GPrT. In that case, we map it to that leaf category, where cosine similarity of mean word embeddings (Mikolov et al., 2013) of the product category and the hierarchy's last two entities is the highest. This choice gives qualitatively good mapping when measured using human evaluation.

**Triplet Count in Customer Support** Note that we do not double the number of triplets by swapping anchor and positive in case of Customer Support, as doing so leads to very similar results, while taking double the compute. This is because in case of Customer Support, E-Manuals are written based on the product

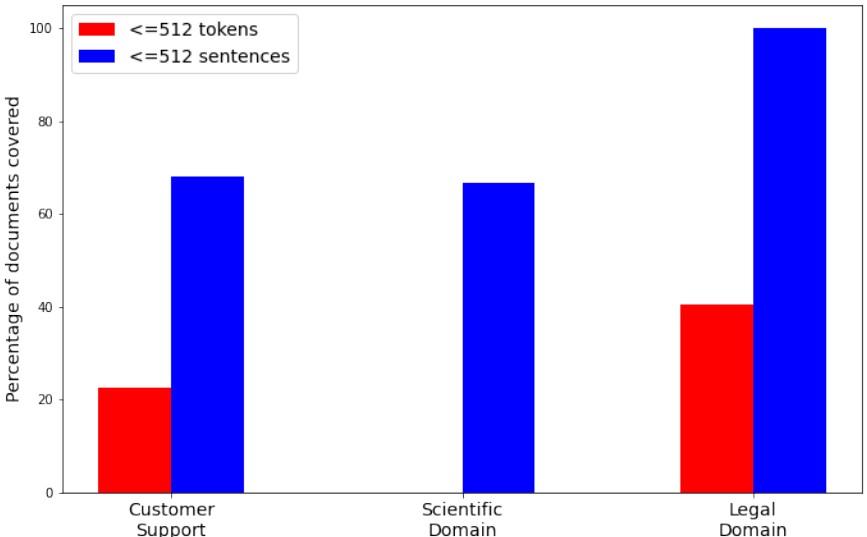

Figure 3: Percentage of documents encoded entirely by RoBERTa-BASE encoder when the input is 512 tokens vs. 512 sentences (The red bar of "Scientific Domain" has a negligible height, and is hence, not visible.)

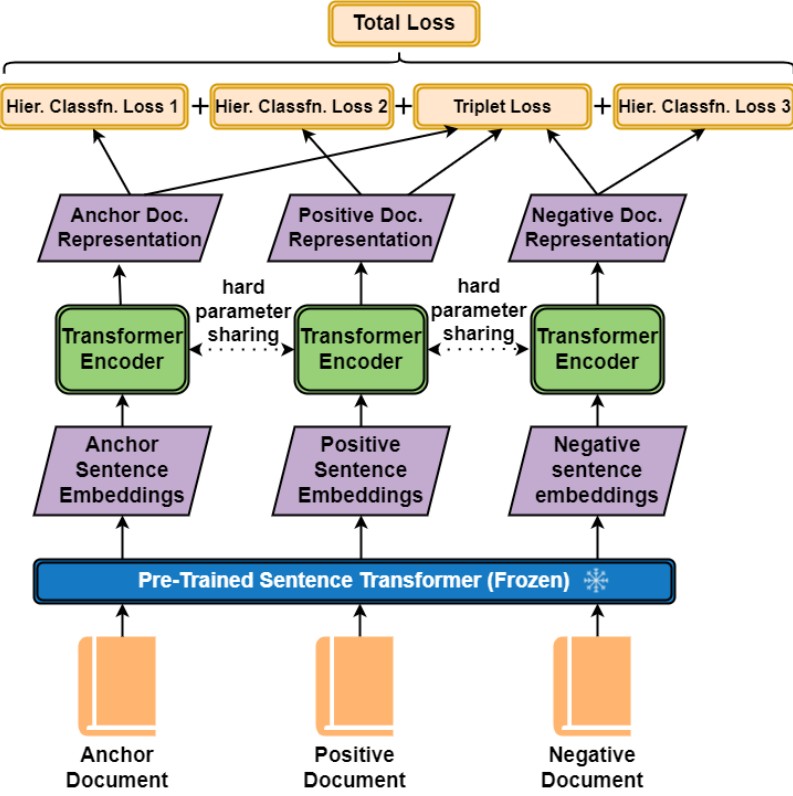

Figure 4: Depiction of **FastDoc**. Anchor, Positive, and Negative Documents are encoded using a Sentence Transformer, followed by a transformer encoder, to give document representations. A combination of Triplet and Hierarchical Classification Losses is used to get the Total Loss

category metadata, while the metadata for Scientific and Legal Domains are assigned after the document is

| Product Category | Hierarchical categories assigned to the product |
|---|---|
| blu-ray player | Electronics > Audio > Audio Accessories > MP3 Player Accessories > MP3 Player & Mobile Phone Accessory Sets |
| stereo equalizer | **Electronics** > **Audio** > Audio Players & Recorders > Stereo Systems |
| laptop docking station | **Electronics** > Electronics Accessories > Computer Accessories > Laptop Docking Stations |
| hot beverage maker | Home & Garden > Kitchen & Dining > Kitchen Appliance Accessories > Coffee Maker & Espresso Machine Accessories > Coffee Maker & Espresso Machine Replacement Parts |

Table 14: Examples of product categories and their corresponding hierarchical categories assigned with the help of the Google Product Taxonomy. Similar products have similar hierarchies

written. Thus, the product category in Customer Support is much more precise and well-defined compared to primary category and overlap in EUROVOC Concepts used in Scientific and Legal Domains respectively, thus requiring lesser data.

## D Downstream Datasets/Tasks

**Datasets used in our work**

Table 15 lists all the downstream datasets used in our work, along with their corresponding tasks and domains.

| Domain | Task | Dataset |
|---|---|---|
| Customer Support | single-span QA | TechQA |
| | multi-span QA | S10 QA Dataset |
| Scientific Domain | NER | BC5CDR JNLPBA NCBI-D |
| | Relation Classification | ChemProt SciERC |
| | Text Classification | SciCite |
| Legal Domain | Contract Review (Span-based Clause Extraction) | CUAD |

Table 15: List of all the datasets along with their corresponding tasks and domains.

### D.1 Customer Support

**S10 Question Answering Dataset.** The S10 QA Dataset (Nandy et al., 2021) consists of 904 question-answer pairs curated from the Samsung S10 Smartphone E-Manual[13], along with additional information on

---

[13]https://bit.ly/36bqs5E

the section of the E-Manual containing the answer. However, the answer might not be a continuous span, i.e., the answer may be present in the form of non-contiguous sentences of a section. The tasks of section and answer retrieval are performed. The dataset is divided in the ratio of 7:2:1 into training, validation, and test sets, respectively.

### D.2   Scientific Domain

### D.3   Legal Domain

The dataset is split 80/20 into train/test, with a small validation set for the preliminary experiments to perform hyperparameter grid search.

## E   Experiments and Results

Here we report certain extra experiments which could not be accommodated in the main paper due to want of space. We also report **additional ablation analysis** on the TechQA Dataset below.

### E.1   Customer Support Domain

**Baselines:**   Details on SPECTER - When pre-training on E-Manuals, instead of initializing the encoder with SciBERT (Beltagy et al., 2019) (as in Cohan et al. (2020)), we initialize the model using EManuals$_{\text{BERT}}$ (Nandy et al., 2021), sample about the same number of E-Manual triplets stated in Cohan et al. (2020) as inputs (using product category information), and use the first 512 tokens per input E-Manual.

**Fine-tuning Setup**

**Fine-tuning on SQuAD 2.0 (Rajpurkar et al., 2018).**   SQuAD 2.0 is a span-based open-domain reading comprehension dataset, consisting of $130,319$ training, $11,873$ dev, and $8,862$ test QA pairs. Before fine-tuning on the task-specific dataset, we fine-tune the encoder on the SQuAD 2.0 training set, as it has been shown to improve the performance on QA tasks (Castelli et al., 2020). The hyperparameters used are the same as mentioned in Rajpurkar et al. (2018).

**Fine-tuning on TechQA Dataset:**   The encoder is fine-tuned on the TechQA Dataset with the same training architecture used when fine-tuning on SQuAD 2.0. Since this is a QA task, a question and one of the candidate technotes separated by a special token is the input. If the technote contains the answer, the target output is the start and end token of the answer, and it is unanswerable otherwise. The hyperparameters used are the ones mentioned in the default implementation[14] of Castelli et al. (2020).

**Fine-tuning on S10 QA Dataset:**   The S10 Dataset is accompanied by 2 sub-tasks - (a) **Section Retrieval** - given the question and top 10 candidate sections retrieved using BM25 IR Method (Nogueira et al., 2019)[15], the task is to find out the section that contains the answer. (b) **Answer Retrieval** - Given a question and the relevant E-Manual section, the task is to retrieve the answer to the question. For section retrieval, a (question, candidate section) pair separated by special tokens ('[CLS]' and '[SEP]' in case of BERT (Devlin et al., 2019) and '' and '' in case of RoBERTa (Liu et al., 2019)) is input to the model, and the ground truth is 0/1 depending on whether the section contains the answer or not. Similarly, in the case of answer retrieval, the question paired with a sentence of the section containing the answer (separated by special tokens) is the input to the model, and the ground truth is 0/1 depending on whether the sentence is a part of the answer or not. In both sub-tasks, the classification token's ('[CLS]' or '') encoder output is fed to a linear layer (followed by Softmax function) to get a probability value. Fine-tuning on each of the sub-tasks yields separate models which are used during inference time for completion of the respective task.

---

[14]https://github.com/IBM/techqa - Apache-2.0 License
[15]BM25 is better than TF-IDF used in Nandy et al. (2021)

| | F1 | HA_F1@1 | HA_F1@5 |
|---|---|---|---|
| $BERT_{BASE}$ | 8.63 | 16.72 | 22.52 |
| $RoBERTa_{BASE}$ | 13.98 | 27.1 | 43.02 |
| Longformer | 15.39 | 29.82 | 42 |
| $EManuals_{BERT}$ | 10.1 | 19.56 | 29.87 |
| $EManuals_{RoBERTa}$ | 13.62 | 26.38 | 38.67 |
| DeCLUTR | 12.52 | 24.26 | 29.59 |
| ConSERT | 10.78 | 20.88 | 31.55 |
| SPECTER | 0.69 | 1.34 | 7.24 |
| $\textbf{\textit{FastDoc}}(Cus.)_{BERT}(hier.)$ | 9.12 | 17.68 | 26.52 |
| $\textbf{\textit{FastDoc}}(Cus.)_{BERT}(triplet)$ | 10.76 | 20.84 | 31.81 |
| $\textbf{\textit{FastDoc}}(Cus.)_{BERT}$ | 7.8 | 15.11 | 24.78 |
| $\textbf{\textit{FastDoc}}(Cus.)_{RoBERTa}(hier.)$ | 13.83 | 26.8 | 37.84 |
| $\textbf{\textit{FastDoc}}(Cus.)_{RoBERTa}(triplet)$ | 12.93 | 25.06 | 40.84 |
| $\textbf{\textit{FastDoc}}(Cus.)_{RoBERTa}$ | 14.89 | 28.85 | 39.04 |

Table 16: Results for the QA downstream task on the TechQA Dataset, **without intermediate SQuAD 2.0 fine-tuning** (Values in red/green indicate if the values are less than/greater than the values got using intermediate SQuAD 2.0 fine-tuning)

For all the fine-tuning experiments on S10 QA Dataset, we use a batch size of 16 (except for the pre-trained DeCLUTR model with DistilRoBERTa$_{BASE}$ backbone, where a batch size of 32 is used), and train for 4 epochs with an AdamW optimizer (Loshchilov & Hutter, 2018) and an initial learning rate of $4 \times 10^{-5}$, that decays linearly.

**Results**

**Performance on TechQA Dataset**

**Analyzing impact of fine-tuning on SQuAD 2.0:** Table 16 shows the results on the TechQA Dataset without intermediate fine-tuning on SQuAD 2.0 Dataset. Intermediate SQuAD 2.0 fine-tuning definitively improves results for 6 out of 8 baselines, and all the **FastDoc** variants.

**Additional Ablation Analysis**

We perform the following ablations on $\textbf{\textit{FastDoc}}(Cus.)_{RoBERTa}$ - (1) We pre-train both the lower and higher-level encoders and fine-tune the lower encoder. This is referred to as $\textbf{\textit{FastDoc}}(Cus.)_{RoBERTa}(FULL)$ (2) We replace the lower encoder sRoBERTa (sentence transformer) with RoBERTa-BASE (still keeping its weights frozen) and refer to it as $\textbf{\textit{FastDoc}}(Cus.)_{RoBERTa}(lower - RoBERTa)$.

| | F1 | HA_F1@1 | HA_F1@5 |
|---|---|---|---|
| $\textbf{\textit{FastDoc}}(Cus.)_{RoBERTa}(FULL)$ | 17.4 | 33.71 | **46.23** |
| $\textbf{\textit{FastDoc}}(Cus.)_{RoBERTa}(lower - RoBERTa)$ | 15.76 | 30.54 | 42.52 |
| $\textbf{\textit{FastDoc}}(Cus.)_{RoBERTa}$ | **17.52** | **33.94** | 44.96 |

Table 17: Additional Ablation Analysis on TechQA Dataset.

Table 17 shows the results corresponding to the ablations and **FastDoc** on TechQA Dataset. We can see that **FastDoc** performs better than $\textbf{\textit{FastDoc}}(Cus.)_{RoBERTa}(lower - RoBERTa)$ on all 3 metrics, and gives better F1 and HA_F1@1 than $\textbf{\textit{FastDoc}}(Cus.)_{RoBERTa}(FULL)$.

**An alternative to the Triplet Loss:** We also observe the effect of using an alternative for triplet loss. We use "quadruplet loss", which breaks similarity into 3 categories instead of the binary notion in triplet loss. We sample 4 documents per input - Anchor(A), near positive(NP), far positive(FP), negative(N). NP is most similar to anchor, followed by FP, and N. E.g. in Customer Support, NP has same brand, category as that of anchor, FP has same category, different brand, and N has neither same brand nor category. Quadruplet

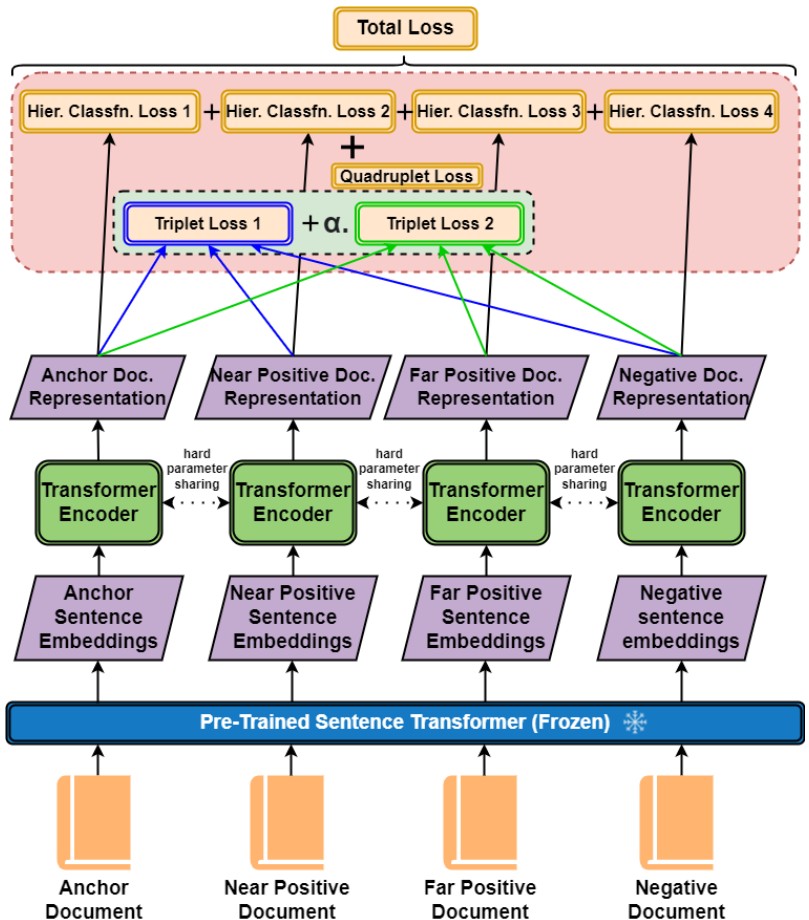

Figure 5: **_FastDoc_**$(Q)$ Pre-training Architecture - It is similar to that of **_FastDoc_**. Differences are - (1) Instead of using a Triplet Loss Function (as in **_FastDoc_**), a Quadruplet Loss Function is used in this case. (2) Anchor, Near Positive, Far Positive, and Negative Documents are taken as inputs.

Loss, denoted by $\mathcal{L}_t$, can be mathematically stated as

$$\mathcal{L}_q = \mathcal{L}_t(A, NP, N) + K.\mathcal{L}_t(A, FP, N) \tag{3}$$

, where $K(0 < K < 1)$ is a constant to reduce weight of the second loss term in Equation 3, as distance between A and NP is to be reduced more than that between A and FP. We denote this variation of **_FastDoc_** as **_FastDoc_**$(Q)$, as depicted in Figure 5. From Table 18, we observe that **_FastDoc_** performs better than **_FastDoc_**$(Q)$ for $K = 0.1, 0.5$ w.r.t all 3 metrics on the TechQA Dataset.

|  | F1 | HA_F1@1 | HA_F1@5 |
|---|---|---|---|
| **_FastDoc_**$(Cus.)_{RoBERTa}(Q, K = 0.1)$ | 16.1 | 31.2 | 43.04 |
| **_FastDoc_**$(Cus.)_{RoBERTa}(Q, K = 0.5)$ | 15.72 | 30.45 | 40.42 |
| **_FastDoc_**$(Cus.)_{RoBERTa}$ | **17.52** | **33.94** | **44.96** |

Table 18: Performance of **_FastDoc_**$(Q)$ vs. **_FastDoc_** on the TechQA Dataset

**Performance on the S10 QA Dataset**

Table 19 shows the performance of baselines and proposed variants on Section Retrieval and Answer Retrieval tasks on S10 QA Dataset. Results are reported on the test set, similar to Nandy et al. (2021). For **Section**

**Retrieval** we report HITS@K - the percentage of questions for which, the section containing the ground truth answer is one of the top $K$ retrieved sections. We report values for $K = 1, 3$. In **Answer Retrieval** a single answer is retrieved, hence HA_F1@1 is reported. The other metrics reported are (a). ROUGE-L [16] (Lin, 2004), and (b). Sentence and Word Mover Similarity (S+WMS) [17] (Clark et al., 2019)[18].

We draw the following inferences from Table 19 - (1) Longformer does not perform well, as global attention does not help in learning local contexts required for answer retrieval. (2) Among the baselines, EManuals$_{\text{RoBERTa}}$ gives the best HITS@1, RoBERTa$_{\text{BASE}}$ gives the best ROUGE-L F1, and DeCLUTR gives the best S+WMS score and HA_F1@1. This shows that MLM and span-based contrastive learning help extract non-contiguous answer spans. (3) Similar to TechQA, **_FastDoc(Cus.)$_{RoBERTa}$_** variants perform better than **_FastDoc(Cus.)$_{BERT}$_** variants. (4) Compared to TechQA, HA_F1@1 scores on S10 QA Dataset are better, as answering questions from a single device is easier than answering questions from diverse sources. (5) In Section Retrieval, **_FastDoc(Cus.)$_{RoBERTa}$_** gives the best HITS@1, suggesting that, pre-training using document-level supervision helps generalize to a device not seen during pre-training. (6) **_FastDoc(Cus.)$_{RoBERTa}$_** gives the best ROUGE-L F1 and the second-best S+WMS score, while **_FastDoc(Cus.)$_{RoBERTa}$_**$(hier.)$ gives the best S+WMS score, and the second-best ROUGE-L F1 and HA_F1@1, as mapping documents to hierarchical labels during pre-training can generalize to the context of a device E-Manual not seen during pre-training. The hierarchy is particularly robust due to the variety in hierarchical labels. Combining it with triplet loss improves the lexical context, as can be seen from the value of ROUGE-L F1.

| | HITS@K | | Answer Retrieval | | |
|---|---|---|---|---|---|
| | $K = 1$ | $K = 3$ | ROUGE -L F1 | S+ WMS | HA_ F1@1 |
| BERT$_{\text{BASE}}$ | 76.67 | 91.11 | 0.792 | 0.411 | 44.87 |
| RoBERTa$_{\text{BASE}}$ | 80 | **93.33** | 0.812 | 0.454 | 45.39 |
| Longformer | 75.56 | **93.33** | 0.768 | 0.415 | 41.1 |
| EManuals$_{\text{BERT}}$ | 81.11 | **93.33** | 0.8 | 0.429 | 44.25 |
| EManuals$_{\text{RoBERTa}}$ | **82.22** | **93.33** | 0.82 | 0.444 | 44.73 |
| DeCLUTr | 76.67 | 92.22 | 0.818 | 0.455 | **46.71** |
| ConSERT | 78.89 | 92.22 | 0.778 | 0.389 | 40.85 |
| SPECTER | 77.78 | **93.33** | 0.802 | 0.429 | 43.59 |
| **_FastDoc(Cus.)$_{BERT}$_**$(hier.)$ | 81.11 | **93.33** | 0.791 | 0.427 | 43.18 |
| **_FastDoc(Cus.)$_{BERT}$_**$(triplet)$ | 77.78 | **93.33** | 0.798 | 0.419 | 42.93 |
| **_FastDoc(Cus.)$_{BERT}$_** | 78.89 | **93.33** | 0.79 | 0.412 | 41.75 |
| **_FastDoc(Cus.)$_{RoBERTa}$_**$(hier.)$ | 78.89 | 92.22 | 0.82 | **0.478** | 46.69 |
| **_FastDoc(Cus.)$_{RoBERTa}$_**$(triplet)$ | 80 | **93.33** | 0.811 | 0.437 | 43.78 |
| **_FastDoc(Cus.)$_{RoBERTa}$_** | **82.22** | **93.33** | **0.828** | 0.463 | 46.22 |

Table 19: Results on the S10 QA Dataset (Best value for each metric is marked in **bold**, while the second-best value is underlined).

**Qualitative Analysis of answers predicted by a proposed variant and a baseline**

We discuss qualitative results with 2 questions from TechQA Dataset and 1 question from S10 QA Dataset and the answers **_FastDoc$_{RoBERTa}$_** and a consistently well-performing baseline EManuals$_{\text{RoBERTa}}$ [19] provide for each question. These questions, ground truth and predicted answers are listed in Table 20. The first question is a procedural question ('How' type), where both models give extra information w.r.t the ground truth. However, **_FastDoc$_{RoBERTa}$_** performs better in extracting the exact number corresponding to the 'Fix' which EManuals$_{\text{RoBERTa}}$ misses. The second question is in essence two questions together where one

---

[16]used `https://pypi.org/project/py-rouge/`

[17]used `https://github.com/eaclark07/sms`

[18]ROUGE-L F1, S+WMS are reported, as all questions in the S10 QA Dataset are answerable, and these metrics make sense when each question has a ground truth answer.

[19]Note that even though Longformer performs well on TechQA, it does not perform well on S10 QA Dataset. For compactness, we chose only one consistently well performing baseline. However, the illustrations will be similar.

is procedural ('How type') and the other is factual ('Is' type) question. Both the models output short answers that have minimal overlap with the ground truth, suggesting that it is difficult to answer multiple questions of different types at one go. However, $FastDoc_{RoBERTa}$ is nearer to the answer, as it suggests the issue is related to 'WS-Proxies' (mentioned in the ground truth), but that does not appear in the baseline's answer. The third question is a location-based question ('Where' type). $FastDoc_{RoBERTa}$ answers it almost correctly, with just some extra information on the paragraph containing the answer, whereas the baseline confuses 'fingerprint list' for 'fingerprint recognition'. The observations point to the proposition that **$FastDoc_{RoBERTa}$ performs better at extracting numerical entities, tackling multiple questions in a sample, and answering location-based questions.**

| Dataset | Question | Ground Truth Answer | $FastDoc(Cus.)_{RoBERTa}$ | $EManuals_{RoBERTa}$ |
|---|---|---|---|---|
| TechQA Dataset | We are using Portal Server 8.5 and when starting the Portal server we see following error:\n\n0000005e MBeanStarter E com.ibm.wps.scripting.server. MBeanStarter load-Extensions failed to load extension: key: ServerExt.applications / value: com.ibm.wps.scripting.server. ApplicationServerExtensions How can we resolve this error? | A code fix for this issue is integrated into the Web-Sphere\n Portal & WCM Combined Cumulative Fix 05 (PI31198 [http://www-01.ibm.com/support/docview.wss?uid=swg1PI31198]) for Portal 8.5.0.0 available from Fix Central: | A code fix for this issue is integrated into the WebSphere\n Portal & WCM CombinedCumulative Fix 05 (PI31198 [http://www-01.ibm.com/support/docview.wss?uid=swg1PI31198]) for Portal\n 8.5.0.0 available from Fix Central: <LONG URL> | A code fix for this issue is integrated into the WebSphere\n Portal & WCM Combined Cumulative Fix 06 (PI36060 [http://www-01.ibm.com/support/docview.wss?uid=swg1PI36060]) for Portal\n 8.5.0.0 available from Fix Central: <LONG URL> |
| | I have made changes to a WSDL stored in DP local filesystem. How can I make the WSP reflect this change since DataPower Gateway 7.1.0 WebService Proxy WebGUI doesn't offer the \"Refresh\" button anymore? Is there a best practice to refresh the WSP if I modified my WSDL? | In a **WS Proxy**, you can refresh WSDL from an URL using WSDL Cache policy. However, if the WSDL is hosted on the DataPower appliance, such as in local:///, rather than on an external URL, the WSDL Cache Policy does not apply. For instance, changes made to local:///test.wsdl are not picked up by **WS Proxy** .... To refresh a WSDL in the local:/// directory, disable and re-enable the service. | To minimize the impact of this issue, consider creating additional **WS-Proxi**es to reduce the number of WSDLs per proxy. | Add/remove the WSDL from the WSP Gateway each time you change the WSDL. 2) Add one char to the user summary and then backspace it. The apply button will be enabled now. 3) Add a WSDL Cache Policy. |
| S10 QA Dataset | I need the registered fingerprint list. Where can I find this? | From Settings, tap Biometrics and security >Fingerprints for the following options: The list of registered fingerprints is at the top of this list. You can tap a fingerprint to remove or rename it. | Add, delete, and rename fingerprints. From Settings, tap Biometrics and security > Fingerprints for the following options: The list of registered fingerprints is at the top of this list. You can tap a fingerprint to remove or rename it. | Use fingerprint recognition to verify your identity in supported apps and actions. From Settings, tap Biometrics and security >Fingerprints. This is not available on the Galaxy S10e. This is only available on the Galaxy S10e. |

Table 20: Qualitative Analysis of answers predicted by $FastDoc(Cus.)_{RoBERTa}$ and the $EManuals_{RoBERTa}$ baseline on the TechQA and S10 QA Datasets. (The parts in green overlap with the ground truth, and the parts in red do not overlap.)

In an effort to understand the reason behind **FastDoc** being better at extracting numerical entities or answering location-based questions, we see how well are such entities shared by anchor and positive documents compared to anchor and negative documents in Table 21.

| | Number of common entities between anchor and positive | Number of common entities between anchor and negative | Number of common entities in (anchor, positive) and not in negative |
|---|---|---|---|
| Numerical | 20.7 | 10.2 | 13.3 |
| Noun Phrase | 61.5 | 6.6 | 57.9 |

Table 21: Analysis of the distribution of numerical and noun phrase entities across document triplets

We observe that overlap between anchor and positive is more than that between anchor and negative. Also, number of entities in (positive, anchor) and not in negative is more than the number of common entities between anchor and negative. Hence, numerical entities and locations (subset of noun phrases) are shared across highly similar documents, which is captured in the contrastive learning task performed during pre-training **FastDoc**. Hence, **FastDoc** is proficient at answering numerical entity and location-based questions.

### E.2 Scientific Domain

Since recent works have used citations as a similarity signal, we report the performance of **FastDoc** using citations as a similarity signal in Table 22. This gives a satisfactory performance, showing that **FastDoc** works on different metadata types. However, on average, a system using citations does not perform as well as when using "primary category". In line with the recent works on Contrastive Learning, we apply **FastDoc** on triplets sampled using citations as a similarity signal and denote it as $FastDoc(Sci.\text{-}Cit.)_{BERT}$. We can see in Table 22 that on an average, $FastDoc(Sci.)_{BERT}$ performs better than $FastDoc(Sci.\text{-}Cit.)_{BERT}$.

| Field | Task | Dataset | $FastDoc(Sci.\text{-}Cit.)_{BERT}$ | $FastDoc(Sci.)_{BERT}$ |
|---|---|---|---|---|
| BIO | NER | BC5CDR | 87.55 | 87.81 |
| | | JNLPBA | 75.9 | 75.84 |
| | | NCBI-D | 85.12 | 84.33 |
| | REL | ChemProt | 73.8 | 80.48 |
| CS | REL | SciERC | 80.8 | 78.95 |
| Multi | CLS | SciCite | 84.13 | 83.59 |
| | | AVERAGE | 81.22 | **81.83** |

Table 22: $FastDoc(Sci.)_{BERT}$ vs. $FastDoc(Sci.\text{-}Cit.)_{BERT}$ in tasks presented in Beltagy et al. (2019). We report macro F1 for NER (span-level), and for REL and CLS (sentence-level), except for ChemProt, where we report micro F1.

### Comparison with GPT-3.5

We compare **FastDoc** with GPT-3.5 in the zero and one-shot settings for some tasks in the Scientific Domain in Tables 23 and 24 respectively. We can see that our proposed **FastDoc** performs much better compared to the highly capable and much larger GPT-3.5 in both zero and one-shot settings.

| Field | Task | Dataset | $FastDoc(Sci.)_{BERT}$ | GPT-3.5 (Zero-Shot) |
|---|---|---|---|---|
| BIO | NER | BC5CDR | **87.81** | 56.04 (-36.18%) |
| | | JNLPBA | **75.84** | 41.25 (-45.61%) |
| | | NCBI-D | **84.33** | 50.49 (-40.13%) |
| | REL | ChemProt | **80.48** | 34.16 (-57.56%) |

Table 23: Results of **FastDoc** vs. zero-shot GPT-3.5 on some of the tasks presented in Beltagy et al. (2019).

| Field | Task | Dataset | $FastDoc(Sci.)_{BERT}$ | GPT-3.5 (One-Shot) |
|---|---|---|---|---|
| BIO | REL | ChemProt | **80.48** | 48.64 (-39.56%) |

Table 24: Results of **FastDoc** vs. one-shot GPT-3.5 on some of the tasks presented in Beltagy et al. (2019).

### E.3 Legal Domain

# F Pre-training Compute of *FastDoc* relative to the baselines

**Choice of GPU-Hours as a metric for measuring pre-training compute**

*FastDoc* and the baselines in Table 6 use a BERT/RoBERTa backbone, making modules of sharding, data parallelism, etc. uniform across models. Hence, GPU-Hours is appropriate for compute. Also, several works on pre-training such as Devlin et al. (2019); Liu et al. (2019); Cohan et al. (2020); Giorgi et al. (2021) report the number of GPUs and the time needed for pre-training, i.e., they support the metric of GPU-Hours.

Also, we compare the metrics of GPU-Hours and FLOPS (Floating-point operations per second) in Table 25. Note that FLOPS is another reliable metric for measuring computer performance[20], and hence, training compute.

| | FLOPS | GPU-Hours |
|---|---|---|
| EManuals$_{\text{RoBERTa}}$ | $4.46 \times 10^18$ | 980 |
| ***FastDoc***$(Cus.)_{RoBERTa}$ | $2.56 \times 10^15$ | 0.75 |
| Speedup | 1745 | 1307 |

Table 25: Comparison of the speedup obtained using ***FastDoc*** in GPU-Hours vs. FLOPS

We observe that the compute speedup (or reduction) in ***FastDoc*** compared to the EManuals$_{\text{RoBERTa}}$ baseline is very close for the two metrics of GPU-Hours and FLOPS, suggesting that GPU-Hours, like FLOPS, is indeed a reliable metric for measuring pre-training compute.

**Clarification of calculation of pre-training compute of SciBERT**

We would like to present the following evidence accompanied by suitable reasoning in support of the calculated value of the pre-training compute -

1. The blog referred to in Footnote 5 of Beltagy et al. (2019) (`https://timdettmers.com/2018/10/17/tpus-vs-gpus-for-transformers-bert/`), titled "TPUs vs GPUs for Transformers (BERT)", discusses the compute requirements for BERT-LARGE and BERT-BASE using different GPU and TPU configurations and specifications. A line from a section of the blog titled "BERT Training Time Estimate for GPUs" states - "On an 8 GPU machine for V100/RTX 2080 Tis with any software and any parallelization algorithm (PyTorch, TensorFlow) one can expect to train BERT-LARGE in 21 days or 34 days". This does not match the sentence in the footnote, which suggests that it is expected to take 40-70 days for pre-training on an 8 GPU machine. Hence, we believe that the sentence in the footnote does not correspond to BERT-LARGE, rather, it intuitively corresponds to SciBERT. The blog was referred to give the reader an idea of how a comparison of GPU and TPU is made.

2. We must mention here that the TPU versions used in Beltagy et al. (2019) and Devlin et al. (2019) are in all probability different. Beltagy et al. (2019) reports its pre-training time corresponding to TPU v3, whereas Devlin et al. (2019) does not mention the exact version of the Cloud TPU used. Also, according to the Google Cloud TPU Release Notes (`https://cloud.google.com/tpu/docs/release-notes#October_10_2018`) we see that the TPU v3 was first introduced (in beta release) on October 10, 2018. However, the first version of the BERT Paper was added to ArXiv (`https://arxiv.org/abs/1810.04805v1`) on October 11, 2018, just 1 day after the beta release of TPU v3 - indicating that, some earlier version of TPU was used for the pretraining experiments.

---

[20]`https://kb.iu.edu/d/apeq`

# G  Analysis and Ablation

## G.1  Catastrophic Forgetting in open-domain

**Hyperparameters:** For all such experiments, we fine-tune for 10 epochs, with a learning rate of $3 \times 10^{-5}$, input sequence length of 512, and batch size of 32. For a task, the best development set results across all epochs is reported.

## G.2  Parameter-Efficient training

The trainable parameters of the encoder during fine-tuning are the same as that in the domain-specific pre-training stage. Hence, as an ablation, we explore the impact of a reduced number of trainable parameters during pre-training by incorporating Parameter-Efficient Training.

| Field | Task | Dataset | $FastDoc(Sci.)_{BERT}$ | $FastDoc(Sci.)_{BERT}(LoRA)$ |
|---|---|---|---|---|
| BIO | NER | BC5CDR | 87.81 | **88.59 (+0.89%)** |
| | | JNLPBA | **75.84** | 61.24 (-19.25%) |
| | | NCBI-D | **84.33** | 39.57 (-53.08%) |
| | REL | ChemProt | **80.48** | 74.32 (-7.65%) |
| CS | REL | SciERC | **78.95** | 62.19 (-21.23%) |
| Multi | CLS | SciCite | **83.59** | 81.35 (-2.68%) |

Table 26: Results of **FastDoc** trained using LoRA vs. proposed **FastDoc** on tasks presented in Beltagy et al. (2019).

Table 26 shows the results of the parameter-efficient training technique of LoRA (Low-Rank Adaptation) (Hu et al., 2022) to observe the effect of using a reduced number of trainable parameters during continual domain-specific pre-training of **FastDoc** in the Scientific Domain. LoRA is applied on the upper encoder of **FastDoc** during pre-training. **FastDoc** performs significantly better compared to when using LoRA in downstream NER, when there are a large number of classes (as in JNLPBA, which has 11 classes), or when the dataset is extremely imbalanced (as in NCBI-Disease, where 91.72% of the training samples belong to a single class). We attribute this to an insufficient number of trainable parameters when using LoRA during pre-training. Similarly, LoRA performs poorly in Relation and Text Classification. On the contrary, there is a meagre reduction in compute from 1.7 to 1.66 GPU-Hours when using $FastDoc(Sci.)_{BERT}(LoRA)$ instead of $FastDoc(Sci.)_{BERT}$, suggesting that LoRA is not beneficial from a compute perspective as well.

## G.3  Absence of document supervision

| Dataset | SCIBERT | *FastDoc* (3 hier. levels) | *FastDoc* (w/o est. meta, tax. 3 hier. levels) | *FastDoc* (w/o est. meta, tax. 11 hier. levels) |
|---|---|---|---|---|
| BC5CDR | 85.55 | 87.81 | 87.6 | **87.88** |
| JNLPBA | 59.5 | 75.84 | 75.91 | **76.06** |
| NCBI-D | **91.03** | 84.33 | 85.02 | 85.05 |
| ChemProt | 78.55 | **80.48** | 76.9 | 76.6 |
| SciERC | 74.3 | 78.95 | 79.26 | **81.21** |
| SciCite | **84.44** | 83.59 | 83.6 | 83.6 |

Table 27: Results of **FastDoc** on tasks mentioned in Beltagy et al. (2019) in Scientific Domain with and without established domain-specific document metadata and taxonomy, compared to a well-performing baseline

| Model | AUPR | Precision@ 80% Recall |
|---|---|---|
| b+ Contracts Pre-training | 45.2 | 34.1 |
| $\textbf{\textit{FastDoc}}(Leg.)_{RoBERTa}$ (4 hier. levels) | 44.8 (-0.88%) | 34.6 (+1.47%) |
| $\textbf{\textit{FastDoc}}(Leg.)_{RoBERTa}$ (w/o est. meta., tax., 4 hier. levels) | 46.7 (+3.32%) | 38.7 (+13.49%) |
| $\textbf{\textit{FastDoc}}(Leg.)_{RoBERTa}$ (w/o est. meta., tax., 17 hier. levels) | **47.9** **(+5.97%)** | **42** **(+23.17%)** |

Table 28: Results of **FastDoc** on CUAD Dataset in Legal Domain with and without established domain-specific document metadata and taxonomy, compared to a well-performing baseline

### G.4 Why *FastDoc* works: Analysis of the interoperability of embeddings

**Interoperability of pre-trained encoder parameters for input token and sentence embeddings**

We present experiments and observations to support the surprising interoperability of input embeddings, in response to the following research questions -

*Q1. How does* **FastDoc** *learn local context?*

**Nature of pre-training inputs:** We contrast the paragraph-level similarity between similar and dissimilar input documents used during pre-training. Given a pair of E-Manuals (from the Customer Support Domain), each paragraph in the two E-Manuals is converted to a fixed-size vector using Doc2Vec model (Le & Mikolov, 2014) trained on Wikipedia, and the similarity score (cosine) with the most similar paragraph from the other E-Manual is considered as a 'Paragraph Similarity Score'. The distribution of this score across similar and dissimilar document pairs is plotted in Fig. 6.

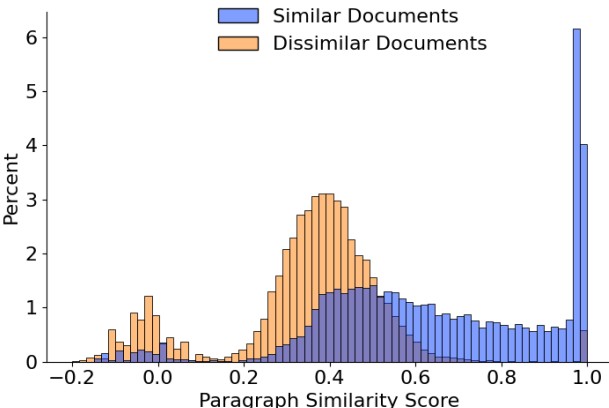

Figure 6: Distribution of 'Paragraph Similarity Score' across similar and dissimilar document pairs

We can infer that paragraph pairs from similar documents are skewed towards higher similarity, with more than half of the samples having a 'Paragraph Similarity Score' > 0.7. The sections where the similarity is even higher are mainly the specific sections where some procedure/task is specified like 'How to calibrate a wireless thermometer?' or the device specifications and/or warnings about using the device/service are given, which would indirectly help the model in the downstream QA task. Similarly, (even the most similar) paragraph pairs in dissimilar documents are skewed towards lower similarity, with a majority of the samples having a 'Paragraph Similarity Score' < 0.5. Thus similar documents have very-similar local (paragraph-level) contexts, suggesting that, using document-level supervision during pre-training also helps in learning local contexts.

**Analysis of Local Context learned using *FastDoc*:** Document-level supervision using *FastDoc* performs well on token-level tasks that require learning from local context (such as NER, relation classification, etc.), even though it was not explicitly trained on learning local context. To validate this, we measure the influence of local context on a random token vis-a-vis a standard MLM model (that learns from local context in accordance with the supervision signal). We randomly sample 500 sentences from each of the 3 domains. For each sentence, we take a random token and calculate the change in its prediction probability on masking other tokens in the sentence. Table 29 shows the Spearman Correlation of this change between two models (***FastDoc*** and a domain-specific model pre-trained using MLM) for each domain. We observe that the correlation is moderately high for all domains, showing that the local contexts is learned by ***FastDoc*** to a reasonable extent.

| Domain | Model using *FastDoc* | Model using MLM | Correlation |
|---|---|---|---|
| Customer Support | $FastDoc(Cus.)_{RoBERTa}$ | $EManuals_{RoBERTa}$ | 0.368 |
| Scientific Domain | $FastDoc(Sci.)_{BERT}$ | SCIBERT | 0.481 |
| Legal Domain | $FastDoc(Leg.)_{RoBERTa}$ | $RoBERTa_{BASE}$ + Contracts Pre-training | 0.393 |

Table 29: Correlation of the change in masked token prediction probability between ***FastDoc*** and MLM, corresponding to other masked tokens, across domains.

*Q2. Are the relative representations preserved across the two embedding spaces?*

**Qualitative Evaluation of Relative Document Representations:** We analyze the relative document representations learnt by the pre-trained encoder in ***FastDoc(Cus.)$_{RoBERTa}$***, for both sentence-level and token-level input embeddings. For 4 different product categories (printer, plumbing product, battery charger, indoor furnishings), we consider 5 E-Manuals each containing between $400 - 512$ tokens so that it complies with the maximum number of tokens accepted by $BERT_{BASE}$ or $RoBERTa_{BASE}$ as inputs. The cosine similarity between the E-Manual representations of each of the $^{20}C_2 = 190$ E-Manual pairs corresponding to both types of input embeddings is obtained, and normalized (using max-min normalization). The similarity values corresponding to the two types of input embeddings are positively correlated to each other, with the Pearson Correlation value being 0.515. We further take the category-wise average representations, and repeat this experiment. We find that the Correlation for the similarity values is 0.977. Hence, the relative representations in both the representation spaces are highly correlated, which justifies good downstream performance when an encoder pre-trained on sentence embedding inputs is fine-tuned on token embedding inputs.

For a visual analysis of these E-manuals, PCA (Principal Component Analysis) is applied over the document representations to reduce the vector dimension from 768 to 2. These representations are then plotted (as shown in Fig. 7) for the pre-trained encoder in ***FastDoc(Cus.)$_{RoBERTa}$*** and two types of input embeddings (sentence and token level). Different product categories are shown in different colors. We infer that independent of whether the inputs are sentence or token-level embeddings, the E-Manuals are clustered in a similar manner across the two representation spaces, and hence, the relative representations are preserved across the two embedding spaces.

*Q3. How are pre-training and fine-tuning compatible?*

**Compatibility between pre-training using *FastDoc* and downstream tasks via few-shot fine-tuning:** To test this compatibility, we fine-tune on a small number of samples in a few-shot setting. We perform 50-shot fine-tuning (i.e., fine-tuning on 50 training samples) on 3 tasks from 3 domains - Span-Based Question Answering on TechQA Dataset from Customer Support (with no intermediate SQuAD Fine-tuning), Text Classification on SciCite Dataset from Scientific Domain, and Span Extraction on CUAD Dataset from Legal Domain. We compared ***FastDoc*** (pre-trained using document-level supervision) with $RoBERTa_{BASE}$ /$BERT_{BASE}$ (pre-trained without any document-level supervision). Tables 30, 31, and 32 show the results, along with the respective improvements when using ***FastDoc***. Better performance of

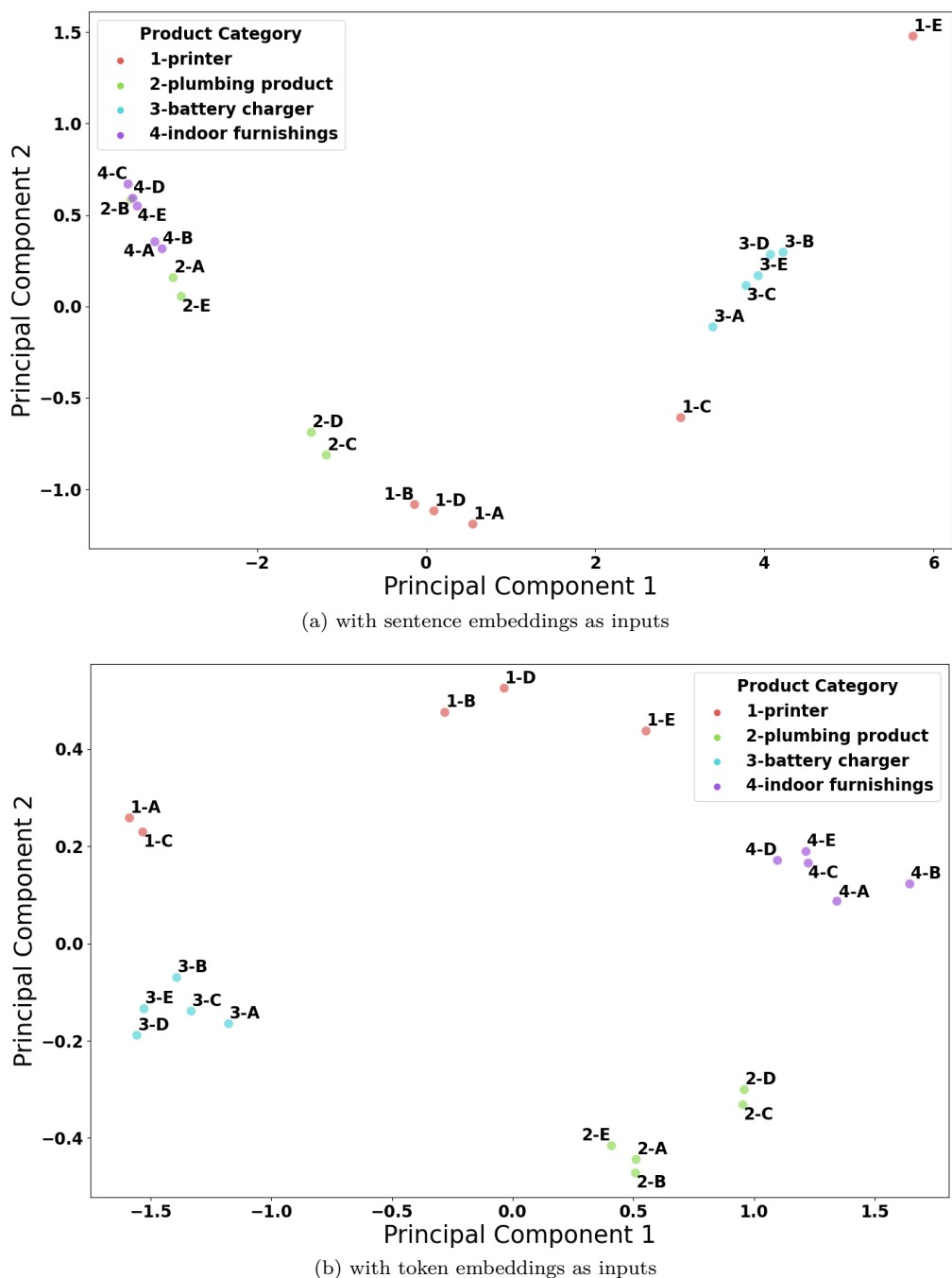

(a) with sentence embeddings as inputs

(b) with token embeddings as inputs

Figure 7: 2D Plots of first two principal components of the document representations of 20 E-Manuals from 4 product categories using **FastDoc(Cus.)**$_{RoBERTa}$ for different types of input embeddings

**FastDoc** across all 3 domains in few-shot fine-tuning setting suggests that (1) Pre-training using document-level supervision is effective across 3 domains (2) **FastDoc** Pre-training and Fine-tuning are compatible.

_Additional Analysis_

**Experiment on analyzing local context similarity of input embeddings:** Fig. 8 shows the distribution of WL (Window Length) corresponding to input sentence and token embeddings for RoBERTa-based **FastDoc(Cus.)**$_{RoBERTa}$ encoder for similar and dis-similar document pairs, for documents that have be-

| Model | F1 | HA_F1@1 | HA_F1@5 |
|---|---|---|---|
| RoBERTa$_{BASE}$ | 0.71 | 1.38 | 2.71 |
| **FastDoc**$(Cus.)_{RoBERTa}$ | **0.86** (**+21.1%**) | **1.66** (**+20.3%**) | **4.85** (**+79%**) |

Table 30: Results on TechQA Dataset in Customer Support Domain

| Model | Macro F1 |
|---|---|
| BERT$_{BASE}$ | 37.75 |
| **FastDoc**$(Sci.)_{BERT}$ | **40.16** (**+6.4%**) |

Table 31: Results on SciCite Dataset in Scientific Domain

| Model | AUPR |
|---|---|
| RoBERTa$_{BASE}$ | 0.13 |
| **FastDoc**$(Leg.)_{RoBERTa}$ | **0.14** (**+7.69%**) |

Table 32: Results on CUAD Dataset in Legal Domain

tween $400 - 512$ tokens. Given a pair of documents, the first being an anchor, WL is 1 more than the distance between an input embedding of the first document, and the most similar input embedding of the second document, averaged across all embeddings of the first document. If similar embeddings are present at nearby positions in two documents, WL will tend to be smaller. Thus WL quantifies the local context similarity of the input embeddings. We observe that the distribution of WL is skewed towards smaller values (i.e., inputs are locally more similar) in similar pairs compared to dis-similar pairs, irrespective of the input (token or sentence) embeddings. Additionally, this also suggests that similar documents inherently induce learning of similarity between sentences and tokens when using **FastDoc**, thus learning from local contexts.

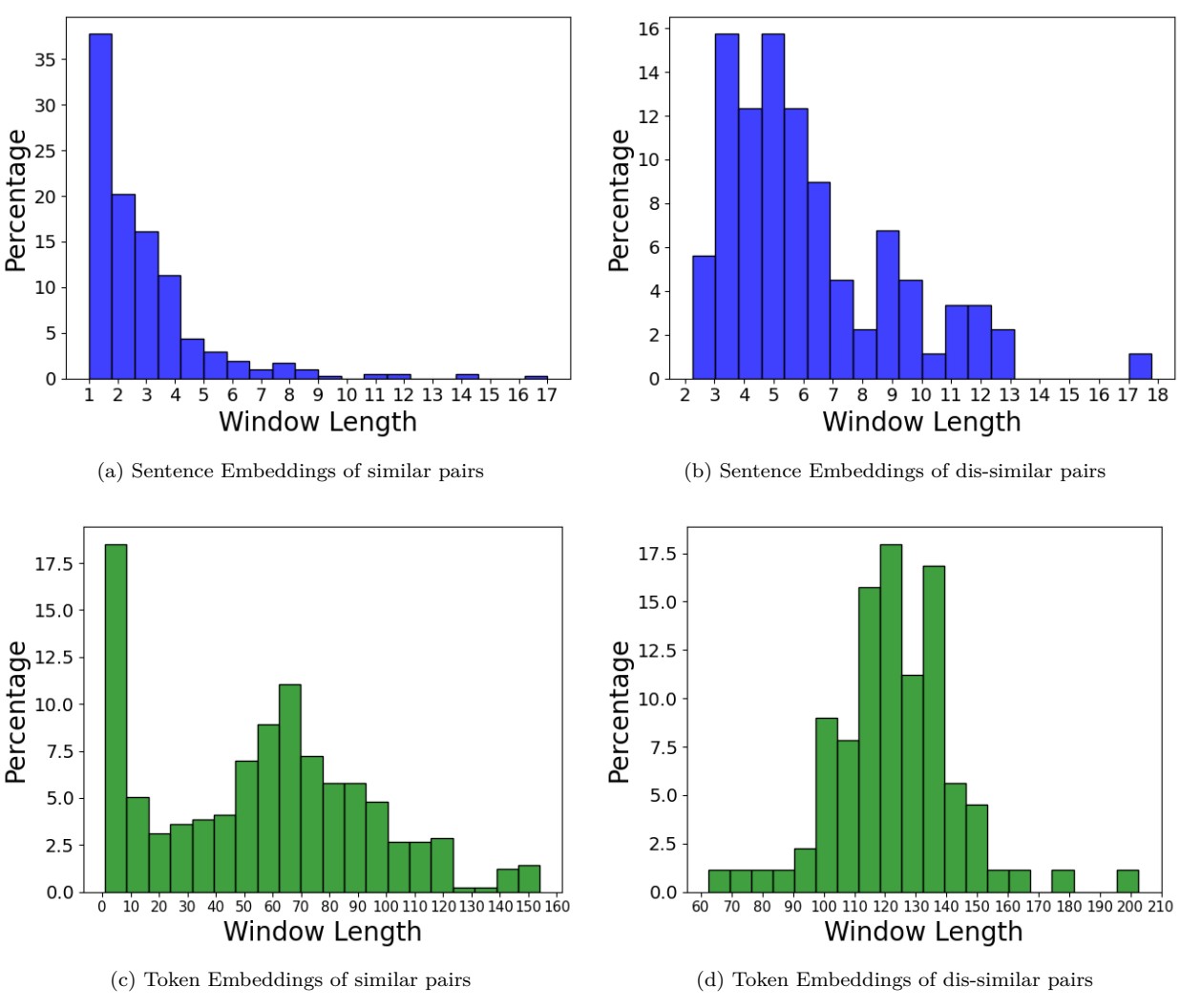

(a) Sentence Embeddings of similar pairs

(b) Sentence Embeddings of dis-similar pairs

(c) Token Embeddings of similar pairs

(d) Token Embeddings of dis-similar pairs

Figure 8: Distribution of WL for sentence and token embeddings as inputs to pre-trained $\textbf{FastDoc}(Cus.)_{RoBERTa}$ encoder for similar and dis-similar document pairs

