# OpenReview forum: "***FastDoc***: Domain-Specific Fast Continual Pre-training Technique using Document-Level Metadata and Taxonomy"
_TMLR — Accepted by TMLR_

### Review · Reviewer_WGMw · 2024-02-24

**Summary Of Contributions:**

This paper proposes an alternative pre-training mechanism for encoder only models when additional information about the task is present. This additional information is either positive/negative examples or hierarchy of domain specific categories or both. These additional information suggests a different training mechanism than MLM or NSP objectives that are typically used in training encoder or decoder models. Here are more specific contributions of the paper:

**A new training mechanism that incorporates example triplets and domain specific hierarchies of categories into the pre-training of BERT-like encoder models.** Triplets for each domain are generated based on corresponding categories such as product categories. This triplets are used with a margin-based contrastive loss. Hierarchies are taken from the corresponding domains when they are available; otherwise, Google Product Taxonomy of Arxiv Category Taxonomy are used to generate hierarchies. These hierarchies are used in a classification objective.

**A hierarchical architecture for pretraining.** A sentence encoder model, such as ROBERTa with an output aggregation such as mean, is used to generate sentence encodings in a document. This encodings are used as inputs to a higher level BERT model that generates sentence level encodings conditioned on other sentences within a document. The higher level encoder is trained with the contrastive learning and classification objectives while the lower level encoder is frozen. This hierarchical approach with sentence level input encodings extend the context length of the model substantially compared to a token level model. Finally, the higher level transformer blocks are combined with the frozen token level embeddings to fine-tune on a downstream domain.

**Extensive experiments on three different domains.** The paper studies three different domains that share positive/negative splitting of examples and information hierarchies. It shows improvement in all three domains compared to other encoder baselines while training a tiny fraction of the available data -- on customer support domain, it is only 3% of the entire corpus. I think this result nicely explains why rather than increasing the size of a corpus, adding auxiliary information can not only drop the labeling requirement but also give better results.

**Audience:**

Yes

**Broader Impact Concerns:**

There are no additional ethical concerns.

**Claims And Evidence:**

Yes

**Requested Changes:**

Based on the above points that I made, a few changes in the paper are critical.

1. Architectural changes or extra information, which one is more critical? Baselines should also use the extra information for a more fair comparison.

2. Estimating compute using GPU hours is unreliable and inaccurate. A better way that is not dependent on the hardware and implementation is needed.

3. A clarification and additional experiments on the long context aspect is needed.

**Strengths And Weaknesses:**

**Strenghts**

Aligned with the contributions that I highlighted above, the paper (i) proposes an easy way to incorporate new information in the form of triplets and hierarchies, (ii) a new hierarchical model that learns from the new structured information, and (iii) extensively shows that it improves compared to baseline models.

**Weaknesses**

There are several points that requires improvement, clarification/discussion.

*Improvement*
1. One major concern is that baselines seem to not trained with the extra information that FastDoc utilizes. This makes it hard to assess whether the architecture of FastDoc is significant or just the extra information that brings the most benefit. Can you utilize the extra information in pre-training of other baselines to see whether they also improve? Table-5 also lists some examples where it is clear that hierarchy can help but not necessarily a hierarchical model.

2. Another major concern is that GPU hours is an inaccurate way of measuring the compute even when everything about the hardware is the same. It depends a lot on the implementation such as sharding, data parallelism, fast attention, etc. that makes it really hard to compare. Why not use a measure that is hardware independent as in scaling literature (like 6ND) [2] ? Similarly, in Section-F you reference open source blogs to justify your compute results including mapping from TPU hours to GPU hours or understanding the TPU version. This also suggests that hours is not a reliable and accurate measure of compute across a huge range of hardware, implementation, data, etc.

3. Related to the compute concern, $EManuals_{BERT}$ is trained on the whole EManuals corpus (33.3 times larger) than FastDoc but its compute requirement is 1000 times more. Can you explain the reason why?

4. You mention that SPECTER's main problem is due to lack of long context. This suggests that with longer context pre-training, it can achieve better results. Is this a general problem for the domains that you study? If you would use a model that can handle long contexts, can that replace FastDoc? For example, can you try LongT5 [1] for your setting?

5. Can you explain why FastDoc is better at extracting numerical entities or location-based questions? Are these things that are shared across documents that would inform the model via hierarchies or categorical triplets?

6. Does your model benefit from scale -- mostly model size?

*Clarification/Discussion*
1. Taxonomies or metadata are not always available. While the paper extends the results by showing a hierarchical topic modeling can help improve the results, it is not clear if this is applicable to the pre-training in general or would be helpful at all. Should we all be extracting hierarchies for C4 or code corpora?

2. Would this only work for encoder-only models?

3. What-if you had the extra information also during inference time?

4. Section E.3 is empty.

[1] [LongT5: Efficient Text-To-Text Transformer for Long Sequences] (https://aclanthology.org/2022.findings-naacl.55) (Guo et al., 2022)
[2] [Training Compute-Optimal Large Language Models] (Huffman et al, 2022)

---

### Review · Reviewer_Z3uB · 2024-02-26

**Summary Of Contributions:**

The paper proposes a new method to quickly continuously pre-train a LLM towards a specific domain. The key contributions are two-fold.
* A methodology to update a LLM using sentence-embedding input (like hierarchical transformers), but that LLM later on will be used with token embedding input
* A loss based on document similarity labels and hierarchical labels.

**Audience:**

No

**Claims And Evidence:**

No

**Requested Changes:**

I expect the authors clarify those points raised above.

**Strengths And Weaknesses:**

The paper presents two interesting ideas and together (FastDoc) they work well: much lower training cost and as good (and even better) than several baselines in 3 different domains (9 datasets).

My first major concern is that it is unclear about the contribution of using sentence embeddings to the empirical results. What if only the proposed loss is used? We should easily test that by training the baselines with the proposed loss.

My second major concern is that the experiments aren't adequate.
* Why did the authors experiment with only BERT and RoBerta-base? Why not with other models and sizes? This is crucial because: 1) as the authors claim that the proposed method is very fast, experimenting with other LLMs should  be manageable. 2) In some cases, e.g. Roberta-large outperforms Roberta-base + contract-pretraining (see Hendrycks et al 2021, tab 2)
* Why LinkBERT, CDLM (sec 8) weren't used as baselines?
* Numbers for SciBERT on table 3 don't match the numbers in Beltagy et al. Tab 1.

My other concerns are:
* It's important to understand how FastDoc works but unfortunately most information is in Appendix rather than the main text. Still, I found the explanation of how FastDoc in the Appendix G.4 unsatisfying. In Table 22, as the models in comparison are based on the same LLMs, isn't that their distributions should be correlated? Besides, the answer for Q3 doesn't explain why FastDoc performs better than several baselines.
*  In Table 7, on average, FastDoc is even better than Roberta-base on several datasets unrelated to Cus. domain. This is counter-intuitive. How could the Cus. domain help boost the performance of Roberta-base?

---

### Review · Reviewer_hrPo · 2024-02-29

**Summary Of Contributions:**

The paper introduces FastDoc, a novel pre-training framework for NLP that replaces traditional objectives such as masked language modeling and next sentence prediction  and  with document similarity learning and hierarchical classification tasks using domain-specific supervision. Utilizing a hierarchical architecture with lower and higher-level encoders, it significantly reduces pre-training compute requirements (up to 500x) while maintaining or improving performance across domains.

The hierarchical architecture in FastDoc incorporates two levels of encoders to enhance pre-training efficiency. The lower-level encoder is initialized using a pre-trained sentence transformer (sBERT/sRoBERTa) and is fixed during training. The higher-level encoder is initialized with a pre-trained BERT/RoBERTa encoder, operating with sentence embeddings from the lower level. This design choice allows for direct processing of sentence embeddings, enabling the model to capture larger contexts in a single input (e.g. 512 sentences with Bert-base models). Finally, the paper also explores further fine-tuning the higher level network after replacing it with the original token embeddings. The higher-level network is trained with two losses: a contrastive or triplet loss based on document similarity and a supervised loss for classifying documents into a domain-specific taxonomy.

Experiments on various tasks (e.g. QA, classification, NER, etc) are run on 3 domains - scientific, legal, and customer support (e-manuals of products). The proposed model gets moderate improvements over the performance of base-models (BERT-base and Roberta-base)

**Audience:**

Yes

**Claims And Evidence:**

No

**Requested Changes:**

I believe the paper requires a significant round of revision. Addressing or clarifying all six points in the weakness section is necessary for me to consider recommending acceptance."

**Strengths And Weaknesses:**

**Strengths**
1. Current pre-training objectives are very localized and do not consider document context. Therefore, it is worthwhile to explore directions for incorporating document-level similarity.

**Weaknesses**

1. Is it pre-training? - My main concern is about the way the paper is positioned. The document similarity and document hierarchy objectives are supervised objectives for which you need training data. Yes, for some domains, you can derive it via distant supervision (e.g., via arxiv hierarchy such as stat.ML), but that doesn’t mean the document hierarchy information is widely available for all domains. I also think that even though the model is trained via these two objectives and is tested on other tasks, I believe it is not pre-training but transfer learning. I would like to know more about why the authors are positioning this paper as a pre-training paper, as the objectives introduced are not self-supervised at all.

2. Some claims regarding baselines are unfair: The paper asserts that it can drastically reduce pre-training compute (500x) compared to the amount of compute required for pre-training the base models. I am not sure if that's justified, as the models proposed in the paper are built on top of pre-trained models and not trained from scratch. Moreover, claims such as “The amount of data used is negligible compared to the 1.14M Papers used by SCIBERT (Beltagy et al., 2019) during its pre-training.” are also not justified, since SCIBERT used a self-supervised objective and trained the model from scratch. The self-supervised objective allows it to scale, and pre-training from scratch on a new domain makes it necessary to pre-train on a large scale.

3. The paper trains different models for each domain, i.e., one model for customer, one for science, and one for legal. I don't think this is a scalable approach. What is the reason for not training one model that works across all domains?"

4. Catastrophic forgetting experimental setting: I am not sure if the catastrophic forgetting (CF) setting is accurate. Typically, to test CF, we consider an online setting where the model parameters are updated on new tasks, and performance is evaluated on old tasks to see if there is any degradation. In Table 7, the results are on the GLUE dev set, while the fine-tuning was on the train set of GLUE. Why is this considered catastrophic forgetting?

5. For the contrastive loss, I am curious why the max-margin loss is selected instead of the more popular contrastive objective, as seen in eq 1 of the simCLR paper (Chen et al., 2020). Additionally, was a margin of 1 sufficient for all tasks?

6. In Table 1, the papers 'Proximal Policy Optimization Algorithms' and 'Generating Natural Adversarial Examples' are considered as a positive pair. Are they truly similar? In other words, is using the same primary category enough to signal that documents are similar? I would encourage the authors to consider richer signals such as document representations, topic modeling, etc., to determine document similarity.

---

### Decision · Action_Editor_aqXr · 2024-04-21

**Recommendation:** Accept with minor revision

**Comment:**

The paper contains extensive experiments showing that one can continue pre-training of a masked LM backbone (bert, roberta) on three domains using sentence embeddings as input and using document-level similarity losses using document taxonomy information. Despite training the backbone with sentence embeddings as input, experiments show that it can still be further fine-tuned with token embeddings as input for downstream tasks. The proposed document-level training objective is much faster than the masked LM (MLM) objective, but yields better or comparable performance on downstream tasks than continued pre-training at the token level with MLM. The authors also experiment with a heuristic that removes the need for document-level taxonomy information.

Overall, I think this paper is slightly above the bar of acceptance.
I *strongly* suggest authors to include discussions about the limitations in this paper in a limitations section, namely:

- Applicability of the proposed model to decoder-only models
- Applicability of the proposed model when downstream tasks are generation tasks

Reviewers asked also for the following points to be revised that are still not addressed in the rebuttal:

- Tone down (or remove) the claim that the method is faster than SciBert, which starts from scratch, while the method starts from Bert.
- Discuss what is the reason for not training one model that works across all domains.

Please, make sure to incorporate all the feedback from the reviewers in the paper along with the additional experiments performed.

**Audience:**

Interest might be somewhat limited given that the experiments are performed using relatively small BERT/Roberta models, but some findings on the compatibility between sentence embeddings and word embeddings as input to these models might be more broadly interesting.

**Claims And Evidence:**

The paper contains extensive experiments illustrating performance of the proposed method under different settings. Some claims about the speed-ups obtained by the proposed model can be toned down.